# STAGE NET: SPATIO-TEMPORAL ATTENTION-BASED GRAPH ENCODING FOR LEARNING MULTI-AGENT INTERACTIONS IN THE PRESENCE OF HIDDEN AGENTS

## ABSTRACT

Accurate prediction of trajectories for multiple interacting agents following unknown dynamics is crucial in many real-world critical physical and social systems where a group of agents interact with each other, leading to intricate behavior patterns at both the individual and system levels. In many scenarios, trajectory predictions must be performed under partial observations i.e., only a subset of agents are known and observable. Consequently, we can only observe the trajectories of a subset of agents with a sampled interaction graph from a larger topological system while the behaviors of the unobserved agents and their interactions with the observed agents are not known. In this work, we propose STAGE Net, a sequential spatiotemporal attention-based generative model to learn system dynamics with multiple interacting agents where some agents are completely unobserved (hidden) all the time. Our network utilizes the spatiotemporal attention mechanism with neural inter-node messaging to capture high-level behavioral semantics of the multi-agent system. Our analytical results motivate STAGE Net design using spatiotemporal graph with time anchors to effectively model complex multi-agent interactions with unobserved agents and no prior information about interaction graph topology. We evaluate our method on multiagent simulations with spring and charged dynamics and two real-world trajectory datasets. Empirical results illustrate that our method outperforms existing multiagent interaction modeling networks in predicting trajectories of complex multiagent interactions even in the presence of a large number of unobserved agents.

## 1 INTRODUCTION

Understanding the unknown underlying dynamics governing a group of co-evolving agents and how they influence each other's behavior is a crucial task across various domains, including robotics (Mavrogiannis & Knepper (2020), Saha et al. (2020), Abbeel & Ng (2004)), social networks (Alahi et al. (2016a), Luber et al. (2010)), and transportation networks (Jahangiri & Rakha (2015), Wojtusiak et al. (2012), Yu et al. (2015)). It poses a challenge to uncover hidden relations and predict dynamics based on observed trajectories, which is vital for downstream decision-making. An important task in discovering and understanding multi-agent dynamics is predicting the trajectory of all agents over time (trajectory prediction). Deep learning techniques such as latent interaction graphs (Kipf et al. (2018), Alet et al. (2019)), attention-based methods for graphs (Vemula et al. (2017), Hoshen (2017), Kosaraju et al. (2019), Huang et al. (2021)), recurrent neural networks (Rubanova et al. (2019b), Zhan et al. (2019)), and neural message passing (Santoro et al. (2017a), Li et al. (2020)) have been developed to predict emergent behavioral patterns in multi-agent systems. With no explicit information about the underlying interaction dynamics, these models formulate the problem in the form of graph structures, with nodes representing the agents and edges expressing the interaction. These models learn the evolution dynamics of nodes or edge attributes in a self-supervised fashion.

These methodologies assume that the dynamical systems are fully observable, i.e the number of agents in the system is known and the trajectories can be sparsely (Zhu et al. (2021), Huang et al. (2020), Marisca et al. (2022), Sun et al. (2019)) or continuously sampled (Alahi et al. (2016b), Banijamali (2022), Graber & Schwing (2020), Kipf et al. (2018)) However, many applications deal with

unobservable agents due to inherent restrictions in sensing and observation capabilities. For example, in a swarm robotics system, the observable agents might only be the designated leaders, while the others are unobservable due to the lack or malfunctioning of sensors. Alternatively, the external vision sensors used for decision-making might only be capable of perceiving a subset of agents in the system, thus providing a constrained view of the entire system's behavior. Additionally, there might be situations where the presence of additional, hidden agents is unknown, complicating the observation challenge further. A system with such 'hidden agents' will demonstrate a lower number of independent degrees of freedom compared to its true intrinsic dimension. Developing deep learning models that can predict the trajectory of multi-agent systems under the limited observability of agents continues to be a challenging task.

In this paper, we present StageNet, a machine learning model to predict the trajectory of multi-agent systems with unknown (hidden) dynamics and in the presence of unknown (hidden) agents. Our network learns effective spatiotemporal representations conditioned only on observations from visible agents. We leverage the spatiotemporal attention mechanism with neural inter-node messaging to capture high-level behavioral semantics of the multiagent system. Our contributions to this paper are summarized as:

- We develop a multi-agent behavior modeling framework designed for trajectory forecasting. This framework utilizes a dynamic spatiotemporal graph attention mechanism, specifically tailored for systems where only a *subset of agents is observable at any given time*.

- We introduce a dynamic spatiotemporal graph to model structural information across time using observations from visible nodes to recover knowledge representations missing due to unobserved agents

- We employ a graph neural network applied to a spatiotemporal graph to approximate the initial latent posterior distribution. This approach aids in learning representations conducive to the long-term prediction of partially observable systems.

We provide analytical motivation that constructing a spatiotemporal graph from visible nodes in a multi-agent system yields a superior representation of the entire system, subsequently enhancing the performance of visible agent trajectory prediction. Empirically, we demonstrate that our STAGE-Net is capable of learning meaningful representations for multi-agent systems, utilizing four datasets: one with agents exhibiting spring dynamics, another with charged dynamics, a real-world dataset of motion trajectories of joints experiencing sensor failures and a dataset capturing the movement patterns of basketball players. Our model offers improved long-term prediction even when a substantial number of agents are unobservable in these diverse scenarios.

## 2 SPATIAL-TEMPORAL ATTENTION MODEL

### 2.1 PROBLEM DESCRIPTION

We consider a multi-agent system with $M$ homogeneous and heterogeneous agents out of which only $N$ agents could be observed (*Observable Agents*) at any time and the rest $(M - N)$ agents are unobserved (*Hidden Agents*). The number of agents could vary depending on the system and we assume that we do not know the total number of agents and hidden agents present in the system. We could only observe the spatial-temporal state sequences of the observable agents. We model the observable agents as a graph $G = \langle O, R \rangle$ where nodes $O = \{o_1, o_2, o_3, ...o_N\}$ represents the observed agents with $R = \{\langle i, j \rangle\}$ representing the interactions among them. We model the interactions among the agents as graph edges. These functional interactions among agents could be inferred from the physical proximity of the agents or the structure of the system they are placed in. We model the interactions $R = \{\langle i, j \rangle\}$ as a weighted adjacency matrix $A \in \mathbb{R}^{N \times N}$ with $a_{i,j} > 0$ representing an edge going from $i^{th}$ node to the $j^{th}$ with interaction strength given by the value of $a_{i,j}$. For each agent, we denote spatio-temporal sequences as $o_i = \{o_i^t\}$ where $t \in \{t_1, t_2, .....t_T\}$ and $o_i^t \in \mathbb{R}^D$ denote the spatial feature of object $i$ at time $t$. The observation sequences are only available for the observed agents and we have no contextual or state information about the hidden agents. For this work, we assume that observations are temporally aligned and sampled at uniform intervals. We denote the the set of historical state sequence as $\mathcal{X}^H = o_i^{1:T_h}, i = 1, ..., N$ and we aim

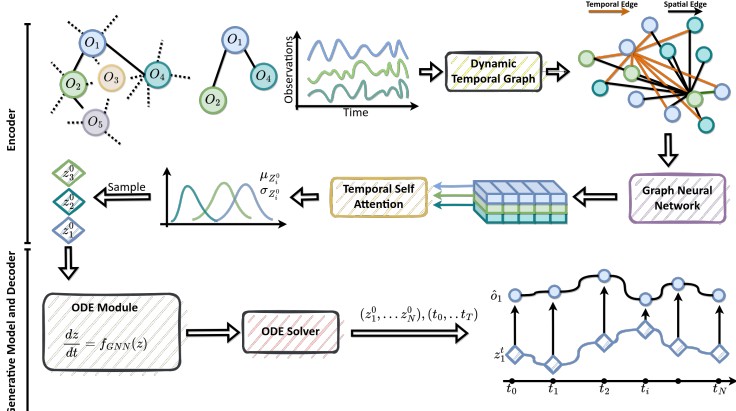

Figure 1: Firstly, the encoder computes the initial latent states for edges and nodes based on the observed sequence of agent observations and adjacency matrix sequence. This computation occurs in two steps: Step 1 involves attention-based representation learning over the dynamic spatiotemporal graph. Step 2 focuses on sequence attention, to learn posterior over the initial latent state. Afterward, the neural ODE framework propagates the latent state through time, and subsequently, the decoder generates predicted observations for the agents. (Best viewed in color)

to estimate $p(\mathcal{X}^{T_h+1:T_h+f}|\mathcal{X}^{1:T_h}, R^{1:T_h})$ to forecast agent trajectories given historical observations up to $t = T_h$ where $T = T_h + T_f$ and $T_f$ denotes the forecasting horizon.

## 2.2 MODEL OVERVIEW

Our method *STAGE Net: Spatial-Temporal Graph Attention Network* is designed to learn representations from spatiotemporal observations of multi-agent systems with interaction graphs sampled from a larger unknown topological system. The *STAGE* learns a parameterized embedding latent space by aggregating temporal representations from all the available multiagent observations weighted by node-dependent attention coefficients. The overall framework is depicted in Figure 1 and it consists of three parts that are trained jointly. (1) An encoder module that maps the observations to latent initial conditions for all the nodes, while taking into account the interactions among entities. (2) A generative neural-ode model characterized by ODE function for latent states for nodes with the goal of learning the latent space dynamics of the system. (3) A decoder that generates the node predictions for the visible agents conditioned on the latent state.

### 2.2.1 DYNAMIC SPATIOTEMPORAL GRAPH WITH TEMPORAL ANCHORS

The core component of STAGE is the dynamics temporal graph to learn and propagate the structural temporal information from observed observations from the sub-graph. Rather than developing an encoder to distill temporal features from the original subgraph (Watters et al. (2017)), our approach constructs a temporal graph derived directly from the agents' observations. A temporal node is instantiated for every $i^{th}$ agent whenever an observation is made at time $t$, and we define a temporal relation, denoted by $r \in R\{\langle i, j \rangle\}$, between agents. Every node in the graph is characterized by a unique feature vector, denoted as $o_{i,t} = [x_{i,t}, v_{i,t}]$, which is concatenation of the agent's spatial location $(x_{i,t})$ and velocity $(v_{i,t})$ for the $i^{th}$ agent at time $t$. Each node is then assigned with time anchors $a_i$ defined as $a_i = t_i - t_{0,i}$ where $t_i$ represents the node's observation time and $t_{0,i}$ is the is the time at which the observation started. This calculated temporal position encapsulates the chronological information, allowing for the nuanced depiction of temporal relationships within the graph. The depiction of temporal relationships is further refined through the construction of edges, based on an edge matrix where each element represents the temporal disparity between two nodes, $i$ and $j$, formalized as $r_{ij} = a_i - a_j$. The existence of an edge and its attributes are contingent upon this time difference, with an edge being formulated and assigned the value of the time difference if it is within a predefined threshold, the maximum allowable gap. Subsequently, we will denote this temporal graph as $\mathcal{G}$.

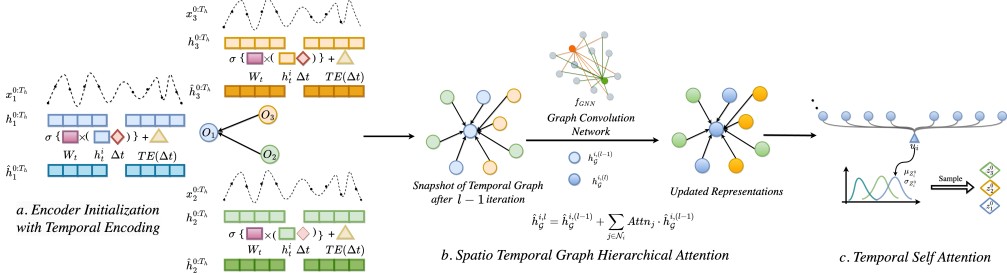

Figure 2: Illustration of the spatiotemporal attention layer in action: On the left side, there's a spatiotemporal graph with each node having an associated time series. In the center(b), you can observe how this layer functions to update the target representation. Finally, the module is passed through the self-attention layer to get the initial latent distribution.

### 2.2.2 TEMPORAL GRAPH HIERARCHICAL ATTENTION

Next, to learn representations for each $i^{th}$ node in the spatiotemporal graph $\mathcal{G}$, we leverage graph attention-based neural message passing to propagate information. The goal of this attention network is to learn aggregated representations from the observations $\mathcal{X}^i_{1:T_h}$ of $i^{th}$ muti-agent and observation set its neighbors $\mathcal{X}^j_{1:T_h}$ where $j \in \mathcal{N}(i)$. We donote the learned representation of $i^{th}$ node in graph at $l^{th}$ layer as $h^{i,(l)}_{\mathcal{G}}$. We initialize the representation encoding with temporal positional encoding $\mathbf{q^i}$ as:

$$h^{i,(0)}_{\mathcal{G}} = \sigma(W_{init}[o_{i,t}\|\Delta t_{start}]) + q^i(\Delta t_{start}) \tag{1}$$

$\sigma(.)$ is a nonlinear activation function and $\|$ is a concatenation operation for tensors. This process is depicted in the left sketch of Figure 2 where this initialization process is shown for a sample graph with three visible nodes. We then update the initialized representations by spatial-temporal attention operations Huang et al. (2021) for each node using graph neural message passing. Similar to Vaswani et al. (2017), we define *query* as the token for which we need a new representation, a *key* as a feature for the source token, and the *value* as the representation or message of the token to be passed. The interaction representation message $\mathbf{Message}_{r \to s} \in \mathbb{R}^{d_h}$ from the $s^{th}$ source node to the $r^{th}$ receiver node is computed as:

$$\mathbf{Message}^{l-1}_{r \to s} = W_v \hat{h}^{s,(l-1)}_{\mathcal{G}}, \quad \hat{h}^{s,(l-1)}_{\mathcal{G}} = \sigma(W_t[h^{s,(l-1)}_{\mathcal{G}}\|\Delta t_{\text{start}}]) + q^i(\Delta t_{\text{start}}) \tag{2}$$

Here, $W_v$ and $W_t$ are linear transformation weight matrices. Next, we find the attention scores for the messages:

$$\mathbf{Attn}^{l-1}_{r \to s} = softmax\{(W_{key}\hat{h}^{s,(l-1)}_{\mathcal{G}})^T(W_{query}h^{r,(l-1)}_{\mathcal{G}}) \cdot \frac{1}{\sqrt{d}}\} \tag{3}$$

Then, all the temporal messages are aggregated to update the node-level context features:

$$h^{r,(l)}_{\mathcal{G}} = h^{r,(l-1)}_{\mathcal{G}} + \sum_{s \in \mathcal{N}_r}(\mathbf{Attn}^{l-1}_{r \to s} \cdot \mathbf{Message}^{l-1}_{r \to s}) \tag{4}$$

This is shown in Figure 2b, where the graph convolution network is used to update the $(l-1)^{th}$ layer's representations.

### 2.2.3 TEMPORAL CONTEXT FEATURE ATTENTION

To embed the above-learned representation to the stochastic latent state, we learn a posterior distribution for each latent state's starting point defined by $z^0_i$. The stochastic latent state is designed to learn the distribution of potential configurations for the visible and hidden agents in the system. In our constructed spatiotemporal graph with time-induced edges, we have $\mathcal{O}(KN)$ nodes where $K$ is the average number of observations for an agent and $N$ is the total number of observed agents. We use a self-attention layer to encode the $K$ observation representations into a single posterior

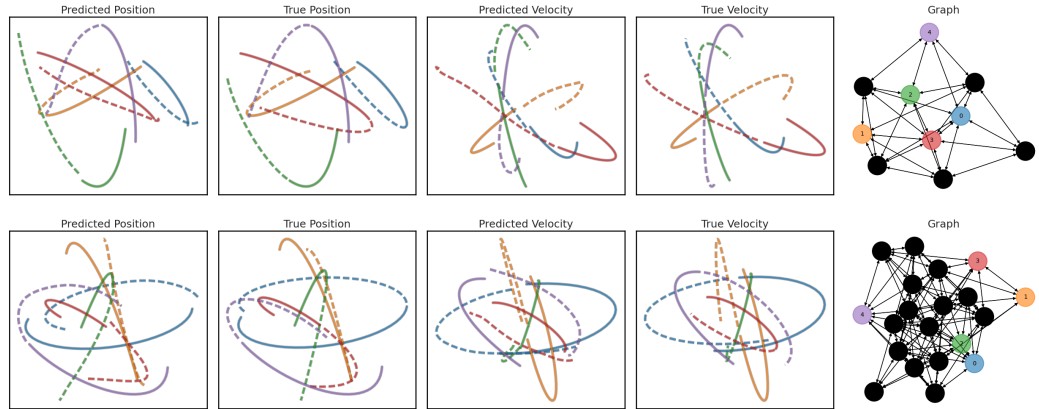

Figure 3: Visualizations depicting predictive trajectories for spring systems involving varying degrees of hidden agents. In the top row, a system with 10 agents and 50% hidden agents is shown, while the bottom row displays a system with 20 agents and 75% hidden agents. Dotted lines represent predicted trajectories, while solid lines represent observed trajectories.

distribution vector for each agent.

$$u_i = \sum_t \mathbf{Attn}(\hat{h}_{\mathcal{G}}^{0,l}, \hat{h}_{\mathcal{G}}^{1,l}, \hat{h}_{\mathcal{G}}^{2,l}...\hat{h}_{\mathcal{G}}^{T_h,l}) \cdot \hat{h}_{\mathcal{G}}^{i,l} \tag{5}$$

here $\hat{h}_G^{i,l}$ is the output of the last graph convolutional layer with temporal encoding followed by nonlinear activation operation similar to 2. The posterior distribution of the initial latent state is then given by:

$$\mu_{z_i}^{t_0}, \sigma_{z_i}^{t_0} = f_{\text{dist}}(u_i), \quad q_\Phi(z_i^{t_0}|\mathcal{X}_{\leq T_h}, \mathcal{G}^{\mathcal{O}}) = \mathcal{N}(\mu_{z_i}^{t_0}, \sigma_{z_i}^{t_0}) \tag{6}$$

### 2.2.4 NEURAL ODE FOR GENERATIVE MODELLING

We model the state evolution of the latent states with a series of ordinary differential equations to drive the latent state forward in time.

$$\dot{z}_i^t = \frac{dz_i^t}{dt} = g_{\text{ode}}(z_1^t, z_2^t, ....z_N^t), \quad z_i^0, ..z_i^T = ODESolver(g_{\text{ode}}, [z_1^0, ..z_N^0], (t_0, ..t_T)) \tag{7}$$

where the $g_{\text{ode}}$ is modeled as a learnable graph convolutional network and models the nonlinear interactions among the agents. An MLP decoder is then employed to reconstruct the trajectory from the latent states.

### 2.2.5 LOSS FUNCTION AND TRAINING

The encoder, decoder, and generative model are trained together by maximizing the evidence lower bound (ELBO), as illustrated below where the first term is the prediction loss for visible nodes, and the second term is the KL divergence.

$$ELBO(\theta, \phi) = \mathbb{E}_{Z^0 \sim q_\phi(Z^0|\mathcal{X})}[\log p_\theta(\mathcal{X})] - \text{KL}[q_\theta(Z^0|\mathcal{X})\|p(Z^0)] \tag{8}$$

### 2.2.6 ANALYTICAL MOTIVATION

Let $G(V(t), E(t))$ be the graph with nodes $V(t)$ and edges $E(t)$ at time $t$. Let $G'$ be a subgraph of $G$ with observed nodes $x_1(t), x_2(t), \ldots, x_N(t)$. The temporal graph $T'$ can be defined as a multiset of the states of graph $G'$ at different time points, represented as: $T' = \{G'(t_1), G'(t_2), \ldots, G'(t_r)\}$ where each $G'(t_i)$ is a member of the multiset representing the state of graph $G'$ at time $t_i$, and additional temporal edges are added between nodes in $G'(t_i)$ and $G'(t_{i+1})$ for all $i = 1, 2, \ldots, r-1$ to represent the temporal connections between the different states of graph $G'$. Here, a multiset is a generalized notion of a set that allows multiple instances of its elements. We first state the following two theorems:

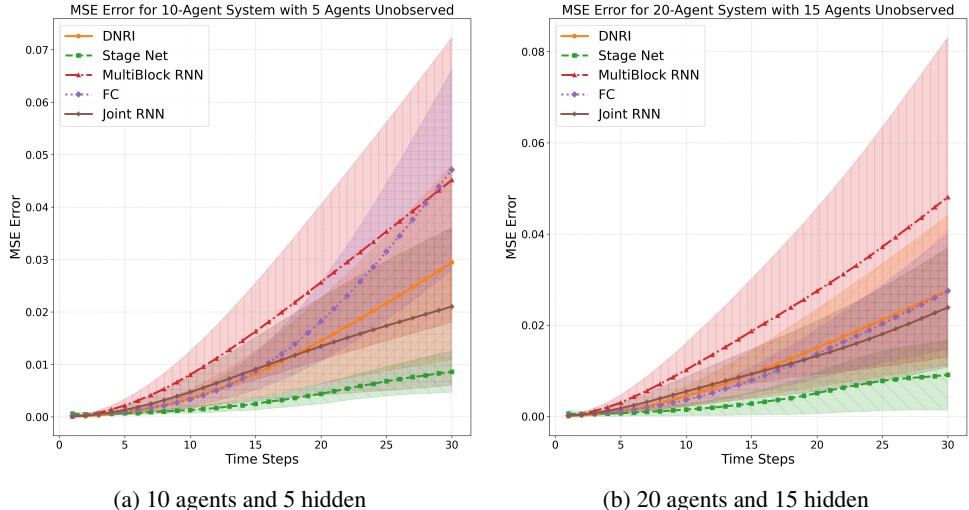

(a) 10 agents and 5 hidden          (b) 20 agents and 15 hidden

Figure 4: MSE Error values vs time for spring systems with 50% and 75% unobservable agents.

**Theorem 1:** *The Fisher information of the embedding of the multiset $X_i$ is greater than the Fisher information of the embedding of each individual element $x_i(t)$ (For proof refer to Supp. Theorem 1)*

**Theorem 2:** *Given the reduced temporal graph $T'$ , the corresponding reduced spatial graph $G'$, and the static spatial graph $G$, if the Fisher information of the embedding of $T'$ exceeds the Fisher information of the embedding of $G'$, i.e., $I(T') > I(G')$ then it follows that the covariance of the reduced temporal graph, $Cov(T')$, is less than the covariance of the reduced spatial graph, $Cov(G')$, represented as: $Cov(T') < Cov(G')$ (For proof refer to Supp. Theorem 2)*

Based on the above two theorems, we can deduce that if $Cov(T')$ and $Cov(G')$ are the estimators of parameters $\theta$ of the full spatial graph $Cov(G)$ then: $Cov(T') < Cov(G')$ i.e. the covariate of the temporal graph $Cov(T')$ is a better estimator of the complete graph $Cov(G)$ than Cov(G'). Hence, constructing a temporal graph from the spatial graph of visible nodes in a multi-agent system where some nodes are unobservable all the time yields a superior representation of the entire system compared to the reduced spatial graph, subsequently enhancing the performance of visible agent trajectory prediction.

## 3   EMPIRICAL EVALUATION

In this section, we evaluate our framework *Stage Net* on two synthetic datasets and two real-world dataset and compare the performance against state-of-the-art methods for trajectory prediction in multi-agent settings. Note that these SoTA methods were all designed assuming all agents are always observed. In subsequent experiments, we address the unobserved agent problem in highly coupled agent settings and assess how prediction performance changes as the percentage of unobserved agents increases.

**Datasets** We validate the effectiveness of our proposed approach by conducting experiments on four distinct datasets: datasets involving agents connected by springs and charged particles (Kipf et al. (2018)), the CMU motion capture dataset (cmu) and the basketball dataset (Yue et al. (2014)). The first two datasets are simulated, where each sample consists of N particles interacting within a 2D box without any external forces. To introduce hidden agents in the simulation, we randomly conceal M agents out of the total N agents in the system after completing all the simulations. As for the motion capture dataset, we specifically select walking sequences from the CMU motion capture dataset. Each sample in this dataset comprises 31 trajectories, where each trajectory corresponds to a single joint of the subject. Similar to the simulated dataset, during both the training and testing phases, we randomly hide joints for the subject. On the other hand, the basketball dataset contains

Table 1: Accuracy Metrics ($\times 10^{-2}$) for $30^{th}$ step in predicting trajectories for simulations with spring interactions.

| Total Agents | Springs 10 | | | | | Springs 20 | | Springs 30 |
|---|---|---|---|---|---|---|---|---|
| Unobserved Agents | 20% | 30% | 40% | 50% | 60% | 75% | 80% | 83.33% |
| Single RNN | $3.20 \pm 1.83$ | $3.88 \pm 2.33$ | $3.85 \pm 2.37$ | $4.51 \pm 2.71$ | $4.33 \pm 2.797$ | $4.81 \pm 3.49$ | $3.61 \pm 2.68$ | $3.60 \pm 2.68$ |
| FC Graph | $6.2 \pm 2.00$ | $5.91 \pm 2.01$ | $5.97 \pm 2.12$ | $5.01 \pm 2.23$ | $4.01 \pm 2.06$ | $2.75 \pm 1.26$ | $2.64 \pm 1.41$ | $2.55 \pm 1.26$ |
| JointRNN | $1.23 \pm 0.96$ | $1.62 \pm 1.20$ | $1.77 \pm 1.28$ | $2.10 \pm 1.50$ | $2.33 \pm 1.73$ | $2.38 \pm 1.30$ | $2.46 \pm 1.67$ | $2.31 \pm 1.48$ |
| D-NRI | $1.49 \pm 0.75$ | $1.85 \pm 0.91$ | $2.34 \pm 1.33$ | $2.49 \pm 1.85$ | $2.30 \pm 1.38$ | $2.77 \pm 1.64$ | $1.97 \pm 1.28$ | $2.06 \pm 1.36$ |
| **STAGE Net (ours)** | $\mathbf{0.20 \pm 0.16}$ | $\mathbf{0.62 \pm 0.23}$ | $\mathbf{0.65 \pm 0.32}$ | $\mathbf{0.78 \pm 0.39}$ | $\mathbf{0.96 \pm 0.58}$ | $\mathbf{0.91 \pm 0.47}$ | $\mathbf{0.96 \pm 0.59}$ | $\mathbf{0.97 \pm 0.51}$ |

Table 2: Accuracy Metrics ($\times 10^{-2}$) for $30^{th}$ step in predicting trajectories for simulations with charged interactions.

| Total Agents | Charged 10 | | | | | Charged 20 | | Charged 30 |
|---|---|---|---|---|---|---|---|---|
| Unobserved Agents | 20% | 30% | 40% | 50% | 60% | 75% | 80% | 83.33% |
| Single RNN | $0.54 \pm 0.48$ | $0.53 \pm 0.49$ | $0.77 \pm 0.54$ | $0.78 \pm 0.63$ | $0.83 \pm 0.69$ | $0.78 \pm 0.54$ | $0.88 \pm 0.65$ | $1.14 \pm 0.73$ |
| FC Graph | $1.17 \pm 0.52$ | $1.01 \pm 0.49$ | $1.21 \pm 0.60$ | $0.91 \pm 0.76$ | $1.49 \pm 0.76$ | $1.65 \pm 0.72$ | $1.71 \pm 0.85$ | $2.33 \pm 1.14$ |
| JointRNN | $0.59 \pm 0.59$ | $0.60 \pm 0.64$ | $0.79 \pm 0.69$ | $0.78 \pm 0.75$ | $0.84 \pm 0.82$ | $0.88 \pm 0.71$ | $1.03 \pm 0.82$ | $1.28 \pm 1.03$ |
| D-NRI | $0.78 \pm 0.49$ | $0.61 \pm 0.49$ | $0.82 \pm 0.51$ | $0.83 \pm 0.60$ | $0.75 \pm 0.62$ | $1.00 \pm 0.66$ | $1.11 \pm 0.85$ | $1.34 \pm 0.93$ |
| **STAGE Net (ours)** | $\mathbf{0.43 \pm 0.42}$ | $\mathbf{0.47 \pm 0.48}$ | $\mathbf{0.59 \pm 0.69}$ | $\mathbf{0.58 \pm 0.65}$ | $\mathbf{0.59 \pm 0.7}$ | $\mathbf{0.72 \pm 0.5}$ | $\mathbf{0.74 \pm 0.72}$ | $\mathbf{0.94 \pm 0.68}$ |

trajectories of five agents out of 10 agents with 50% observability preprocessed into 49 frame data. Additional details about experiment setup and datasets are provided in the appendix A

**Baselines** We compare our network *STAGE* with the following baselines. Since we do not have any existing prior work on this work, we consider state-of-the-art models where a full agent topological graph is known for learning continuous system dynamics. We evaluate against two recurrent neural network (RNN) baselines, Single RNN and Joint RNN, which utilize shared-weight LSTMs for each object and a concatenated LSTM for all objects' states prediction, respectively. We also implement Fully Convolutional Graph Messaging, using a message-passing network decoder similar to (Watters et al. (2017)) over a fully connected graph of visible agents. Furthermore, we consider DNRI (Graber & Schwing (2020)), which combines graph neural networks and variational inference, introducing a latent variable model that captures temporal evolution through an RNN component.

**Experimental Settings** We conducted a series of experiments where different proportions of agents were unobservable. In these experiments, we observed the particles' behavior over a specified time interval, denoted as $[t_0, t_h]$, and expected our model to autonomously learn the dynamics of their interactions. Subsequently, the model was tasked with predicting the trajectories of the particles for the time interval $[t_{h+1}, t_N]$. We utilized a GNN featuring a latent dimension of 64 and incorporated two layers in the temporal graph hierarchical attention module. The temporal context feature attention module was set with an dimension of 128. For solving ODE, we employed the Runge-Kutta solver and we employed a one-layer graph network with a hidden node representation dimension of 128 for the ODE function. For all our simulated datasets and motion dataset experiments, we set the values of $t_h$ and $t_N$ to be 30 and 60, respectively while for the basketball dataset $t_N$ is set to 49. To evaluate the accuracy of our predicted trajectories, we employed the mean squared error (MSE) as the chosen metric.

**Results** Figure 3 displays the qualitative results predicting the spring system's behavior, portraying the model's efficacy in scenarios with 50% and 75% hidden, unobservable agents. Within the graph, nodes colored in black symbolize hidden agents, and those in color represent observable ones. Each

Table 3: Accuracy Metrics ($\times 10^{-2}$) in predicting trajectories for Motion Dataset

| Unobserved Joints | 0 | 5 | 10 | 15 | 20 |
|---|---|---|---|---|---|
| MultiBlock RNN (Schmidt (2019)) | 0.17 | 0.18 | 0.16 | 0.18 | **0.13** |
| FC Graph (Watters et al. (2017)) | 0.14 | **0.13** | 0.19 | 0.16 | 0.16 |
| JointRNN(Schmidt (2019)) | 0.30 | 0.34 | 0.28 | 0.23 | 0.23 |
| D-NRIGraber & Schwing (2020) | 0.16 | 0.30 | 0.17 | 0.30 | 0.20 |
| **STAGE Net (ours)** | **0.11** | **0.13** | **0.11** | **0.10** | **0.13** |

| % Temporal Observability | STAGE Net | DNRI | FC | Joint RNN | Single RNN |
|---|---|---|---|---|---|
| 33% | $1.40 \pm 2.45$ | $2.60 \pm 1.79$ | $2.67 \pm 1.67$ | $1.94 \pm 1.51$ | $1.86 \pm 1.47$ |
| 50% | $1.36 \pm 2.39$ | $2.35 \pm 1.57$ | $2.45 \pm 1.90$ | $1.62 \pm 1.29$ | $1.55 \pm 1.26$ |
| 66% | $1.37 \pm 2.40$ | $2.00 \pm 1.33$ | $2.21 \pm 1.25$ | $1.29 \pm 1.03$ | $1.22 \pm 1.00$ |
| 100% | $1.32 \pm 2.33$ | $2.069 \pm 1.23$ | $1.72 \pm 0.93$ | $1.022 \pm 7.6$ | $0.98 \pm 0.77$ |

Table 4: Accuracy Metrics ($\times 10^{-2}$) for temporal observability for basketball data system with 50% observable agents

Table 5: Ablation study: MSE error for three STAGE net model variants for different configurations for spring dataset.

| Total Agents | Spring 5 | | | | Spring 10 | | | | | | Spring 20 | | Spring 30 | |
|---|---|---|---|---|---|---|---|---|---|---|---|---|---|---|
| Unobserved Agents | 0% | 20% | 40% | 60% | 30% | 40% | 50% | 60% | 70% | 80% | 75% | 80% | 83.33% | 87.33% |
| SN-all connected | 1.2 | 0.6 | 0.45 | 0.49 | 0.67 | 0.76 | 0.93 | 0.63 | 0.58 | 0.67 | 0.58 | 0.60 | 0.81 | 0.59 |
| SN w/o attention | 0.22 | 0.48 | 1.04 | 0.60 | 0.60 | 1.04 | 0.70 | 0.84 | 0.72 | 0.73 | 0.73 | 0.75 | 1.08 | 1.37 |
| SN w/o temporal Encoding | 0.28 | 0.25 | 0.34 | 0.5 | 1.28 | 0.87 | 0.38 | 0.41 | 0.49 | 0.68 | 0.43 | 0.45 | 0.9 | 0.56 |
| STAGE Net original | **0.21** | **0.25** | **0.33** | **0.43** | **0.27** | **0.26** | **0.31** | **0.37** | **0.45** | **0.57** | **0.39** | **0.42** | **0.47** | **0.54** |

SN-all connected: StageNet with visible agents fully connected, SN w/o attention: StageNet without attention mechanism, SN w/o temporal encoding: network with temporal encoding removed, Orignal: network with attention mechanism, temporal encoding and visible graph linkings

agent's observed (represented by solid lines) and predicted trajectories (depicted by dashed lines) are illustrated for both position and velocity. Notably, in the system with 75% unobservable agents, agent number 4 demonstrates a unique case—it maintains no connections with visible agents and is exclusively linked to seven hidden ones. Impressively, even in such a challenging scenario, our model proficiently exploits the spatiotemporal observations of visible agents to predict their trajectories with high accuracy. In Figure 4, a visual representation of the evolution error in dynamics is depicted for the spring system, projecting 30 steps into the future. This specific illustration focuses on scenarios with 50% and 75% unobservable agents. The Stage Net model outperforms all the baseline models in predicting future trajectories while maintaining both low error levels and minimal variance.

Table 1 and Table 2 present the $30^{th}$ step mean-squared error for trajectory prediction in both the spring and charged systems. We conducted experiments on four systems, specifically 5 agents, 10 agents, 20 agents, and 30 agents, respectively. For each system, we gradually hide agents and trained our framework accordingly. STAGE Net consistently outperforms all the baselines for both systems, affirming the efficacy of our framework's design in learning representation. Even when a large portion of the interaction graph is unobserved, our model exhibits minimal prediction errors in experiments involving 20 or 30 agents with only 4 or 5 agents visible. Table 3 shows the prediction results for motion datasets with a different set of joints randomly hidden to train the network. Similar to the spring and charged datasets, our network consistently outperforms the baseline models, demonstrating its superior performance in this context as well. It is noteworthy, however, that in this dataset, baseline models such as RNN and FC Graph exhibit markedly improved performance compared to their counterparts in the spring and charged datasets. This enhanced performance can be attributed to the inherent geometric constraints of joints moving in synchronization with the overall body's trajectory, facilitating more accurate predictions of each joint's trajectory. This contrast is evident when compared to the spring and charged datasets, where an agent's motion is predominantly influenced by its neighboring agents, with no overarching constraints guiding the entire system's movements. Table 4 displays the outcomes of the basketball dataset, where only 50% of the agents are observable. To introduce temporal sparsity, we apply random sparse sampling to encoder observations and utilize them for trajectory prediction, following the methodology outlined in Sun et al. (2019). Our observations reveal that in scenarios involving concealed agents and limited temporal observability in the basketball dataset, STAGE Net surpasses the baseline models in performance.

**Importance of Temporal Encoding and Attention** Our network STAGE Net consists of two main components: the dynamic spatio-temporal graph and the temporal graph attention. The purpose of our study is to gain a deeper understanding of each component within these modules. To do this, we conducted an ablation study where we explored three different variations of the model. In the first variant, we trained the model without any prior knowledge of the relationships between edges in the graph. Consequently, we assumed that all visible agents were interlinked, resulting in the establishment of a fully connected graph during the construction of the temporal graph. StageNet's temporal graph attention module itself comprises two essential elements: the attention module and

the temporal encoding. For the other two variants, we examined models that lacked either attention or temporal encoding. In these variations, we didn't incorporate attention to nodes over time, and we omitted node temporal importance through temporal encoding. We evaluated the performance of these distinct model configurations by quantifying their mean squared error (MSE) across a spectrum of scenarios, encompassing varying numbers of both total agents and hidden agents in the context of spring simulations. Table 5 illustrates the mean squared error (MSE) for different configurations of total agents and hidden agents for spring simulations. Our original model consistently outperformed all alternative model variations across the entire spectrum of scenarios we examined.

**Influence of Hidden Agent on Visible Agent Predictions**

To deepen our understanding of how hidden agents impact the predictions of visible agent dynamics, we conduct tests using models on the spring dataset. We maintain 50% observability and vary the interactions of the hidden agents. For this assessment, we establish connections among all visible agents, thereby forming a fully connected subgraph comprised solely of visible agents. Subsequently, we incrementally augment the number of edges between hidden and visible agents, ranging from $r = 1$ to $r = 5$. Here, a value of $r$ signifies that each hidden agent in the system is connected to $r$ visible agents. Notably, there are no interconnections between any two hidden agents. The illustration of this configuration is provided in Figure 5.

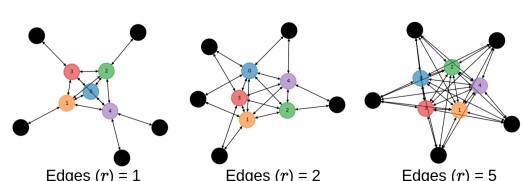

Figure 5: Illustration of the graph configurations to study the influence of hidden agents on visible agent predictions

Figure 6 illustrates the prediction error for the STAGE Net and baseline models. It is evident that as the number of connections between hidden and visible agents increases from 2 to 5, STAGE Net consistently outperforms, maintaining minimal prediction error and variance. In contrast, the baseline models exhibit a decline in predictive accuracy as the number of hidden-visible agent edges increases. Interestingly, when $r = 1$—signifying that each hidden agent is connected to only one visible agent, the

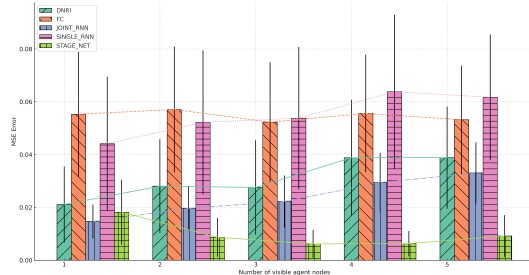

Figure 6: MSE error for models as the number of connections hidden to visible connections are increased.

observed error is higher compared to scenarios where each hidden agent is connected to two or more visible agents. This can be attributed to the absence of hidden agents between any two visible agents, resulting in a betweenness centrality of zero for all visible agent pairs with respect to a hidden agent. In contrast, for other configurations, at least one hidden agent exists between any pair of visible agents. This structural difference enables STAGE Net to adeptly uncover hidden influences through representation learning on spatiotemporal graphs. For additional insights and ablation studies, please refer to Appendix B.

## 4 CONCLUSION

In this work, we have presented a framework for integrating spatiotemporal information from multi-agent observations with multiple co-evolving and interacting agents unobserved. In order to capture the underlying hidden representations of the evolution of dynamics, we propose a dynamic temporal graph to encode the observations to a latent manifold and use a neural ode to propagate the latent interaction dynamics forward. In the future, we would like to estimate the dynamics and intrinsic dimensions of the unobservable agents in the system. We would also like to consider large-scale interacting systems with heterogeneous agents where the interaction relations dynamically evolve over time. While this paper focuses on prediction tasks, an exciting future direction could involve controlling multi-agent systems with hidden agents.

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

## Supplementary

## A EXPERIMENTAL SETUP

### A.1 PROBLEM STATEMENT

We have systematically classified various multi-agent observation scenarios, as outlined in table 6, to position our work within the broader research domain. In our paper, we delve into a particularly challenging scenario, dealing with unobservable agents due to inherent sensing and observation constraints, leading to a system with fewer independent degrees of freedom than its intrinsic dimension. This problem, while seemingly specific, represents a critical and complex challenge within the realm of multi-agent systems. Most prior research in this domain, as summarized in our classification, assumes full observability of agents, whether the sampling is sparse or continuous. Our work, however, tackles a more intricate scenario where some agents are inherently unobservable.

### A.2 DATASET

**Simulated Datasets:** In our particle simulation experiments, we consider N particles, with N taking values from the set $\{5, 40\}$, placed within a 2D box. In the springs model, we randomly establish connections between pairs of particles with a 50% probability and these particles interact via

| Scenario | Description of Problem | References |
|---|---|---|
| Complete observability with known interaction topology | Multi-agent systems where all agents are observable at all times, with a known interaction topology, facilitating the modeling process. | Watters et al. (2017) |
| Complete observability with unknown interaction topology | All agents are observable at all times; however, the interaction topology is not predefined and must be inferred from observational data. | Alahi et al. (2016b) Banijamali (2022) Graber & Schwing (2020) Kipf et al. (2018) Alet et al. (2019) van Steenkiste et al. (2018) Santoro et al. (2017b) |
| Irregular sampling of observations or temporally sparse data | All agents are observable but the observation events are sporadic or irregular, leading to temporal data sparsity. | Rubanova et al. (2019a) Zhu et al. (2021) Huang et al. (2020)Marisca et al. (2022) Sun et al. (2019) |
| Only few agents observable with sparse temporal sampling and unknown interaction topology | Not all agents are observable, with some never being observed, coupled with sparse temporal data collection. | (Ours) |

Table 6: Systematic classification of observation scenarios in multi-agent systems.

Hooke's law, where the force $F_{ij}$ acting on particle $v_i$ due to particle $v_j$ follows Hooke's law: $F_{ij} = -k(r_i - r_j)$, with $k$ as the spring constant and $r_i$ representing the 2D position vector of particle $v_i$. We sample initial positions from a Gaussian distribution ($N(0, 0.5)$), and initial velocities are assigned as random vectors with a norm of 0.5. Trajectories are simulated by numerically solving Newton's equations of motion using a leapfrog integration method similar to Kipf et al. (2018) with a fixed step size of 0.001, and we subsample the trajectories by selecting every 100th step for training and testing.

In contrast, for the charged particle model, we equip each particle with positive or negative charges, $q_i$, sampled uniformly from $\pm q$. The interaction between these charged particles is governed by Coulomb forces, defined as $F_{ij} = C \cdot \text{sign}(q_i \cdot q_j) \cdot \frac{(\mathbf{r}_i - \mathbf{r}_j)}{|\mathbf{r}_i - \mathbf{r}_j|^3}$, where $C$ is a constant. Unlike the springs model, all pairs of charged particles interact, potentially resulting in attraction or repulsion, depending on their relative distances. For each of the simulated datasets, 10,000 training samples and 2,000 testing samples are generated. To incorporate hidden agents within the simulation, we randomly select M agents from the system to hide after the completion of all simulations while only preserving the edges with visible agents.

**CMU Motion Capture Dataset:** The Carnegie Mellon University (CMU) Motion Capture dataset (cmu), a comprehensive and widely recognized collection of motion capture data, was utilized in this study. This dataset embodies a diverse array of human movements, encompassing activities from walking and running to more intricate motions such as dancing, recorded from various subjects. Our empirical focus was on Subject 35 and their walking trajectories. The dataset extracted for our study consists of 8,063 frames, each documenting 31 specific points. All attributes, including position and velocity, were normalized to have a maximum absolute value of 1. We trained our models on 30-timestep sequences and subsequently assessed their performance on sequences of equivalent length.

**Basket Ball Dataset:** In the basketball dataset, each trajectory provides detailed information about the 2D positions and velocities of the offensive team, consisting of 5 players. Initially, these trajectories are divided into 49 frames, which collectively capture approximately 8 seconds of gameplay. During the training phase, all models undergo training using the initial 30 frames extracted from the training trajectories. When it comes to evaluation, the models are presented with input data comprised of sampled trajectories from the first 30 frames, and this sampling strategy is adjusted based

on temporal sparsity. Specifically, for a temporal sparsity of 10%, we select 27 observations from the initial 30 observations for each individual player, and subsequently, the models are tasked with predicting the subsequent 19 frames.

## A.3 Baselines

**Recurrent Neural Networks** We implement two recurrent baselines: Single RNN and Joint RNN. The first RNN baseline utilizes separate LSTMs (with shared weights) for each object. The second baseline, labeled as "joint," combines all state vectors by concatenation and feeds them into a single LSTM, which is trained to predict all future states simultaneously.

**Fully Convolutional Graph Messaging(Watters et al. (2017))** We implement a message-passing network decoder similar to Kipf et al. (2018) operating over a fully connected graph of visible agents with only one edge type.

**DNRI(Graber & Schwing (2020))** DNRI combines the power of graph neural networks and variational inference to model the interactions and dependencies between entities over time. It introduces a latent variable model that captures the temporal evolution of the system by incorporating a recurrent neural network (RNN) component. It allows for inferring the latent variables that represent the hidden states and interactions between the entities at different time steps. By using variational inference, DNRI provides a probabilistic framework that can capture uncertainty and make predictions about future interactions.

Table 7 presents the hyperparameters used for the evaluation of the baselines across all three datasets.

Table 7: List and description of hyperparameters for baselines

| Hyperparameter | Value | Description |
|---|---|---|
| Encoder latent | 128 | Latent size of encoder. |
| Decoder latent | 128 | Latent size of Decoder decoder. |
| Batch_size | 128 | The number of samples processed in a single pass. |
| lr | $5 \times 10^{-4}$ | The learning rate for training the model. |
| Optimizer | Adam | Model optimization algorithm. |
| Teacher forcing steps | 30 | Number of steps for which teacher forcing is applied. |
| Val teacher forcing steps | 30 | Whether to apply teacher forcing during validation. |
| Edge types | 2 | Number of types of edges in the graph. |
| Encoder layers | 2 | Number of layers in the encoder's MLP. |

## A.4 Additional Model Details and Hyperparameters

All components of the Stage Net are illustrated in Figure 7. The hyperparameters utilized to assess Stage Net on all the datasets are listed in Table 8.

**Neural ODE for Generative Modelling** In systems involving continuous multi-variable dynamics, the state's dynamic nature is depicted through continuous values of $t$ over a collection of dependent variables, and it progresses according to a sequence of first-order ordinary differential equations (ODEs):

$$\dot{z}_t^i := \frac{dz_t^i}{dt} = g_i(z_t^1, z_t^2, \ldots, z_t^N)$$

These equations advance the states of the system in tiny steps over time. With the latent initial states $z_0^0, z_0^1, \ldots, z_0^N \in \mathbb{R}^d$ for every object, $z_t^i$ is the resolution to an ODE initial-value problem (IVP) and can be computed at any required times using numerical ODE solvers like Runge-Kutta:

$$z_T^i = z_0^i + \int_0^T g_i(z_t^1, z_t^2, \ldots, z_t^N)dt$$

The function $g_i$ outlines the dynamics of the latent state, and it has been proposed to be parameterized with a neural network in recent research, allowing for data-driven learning.

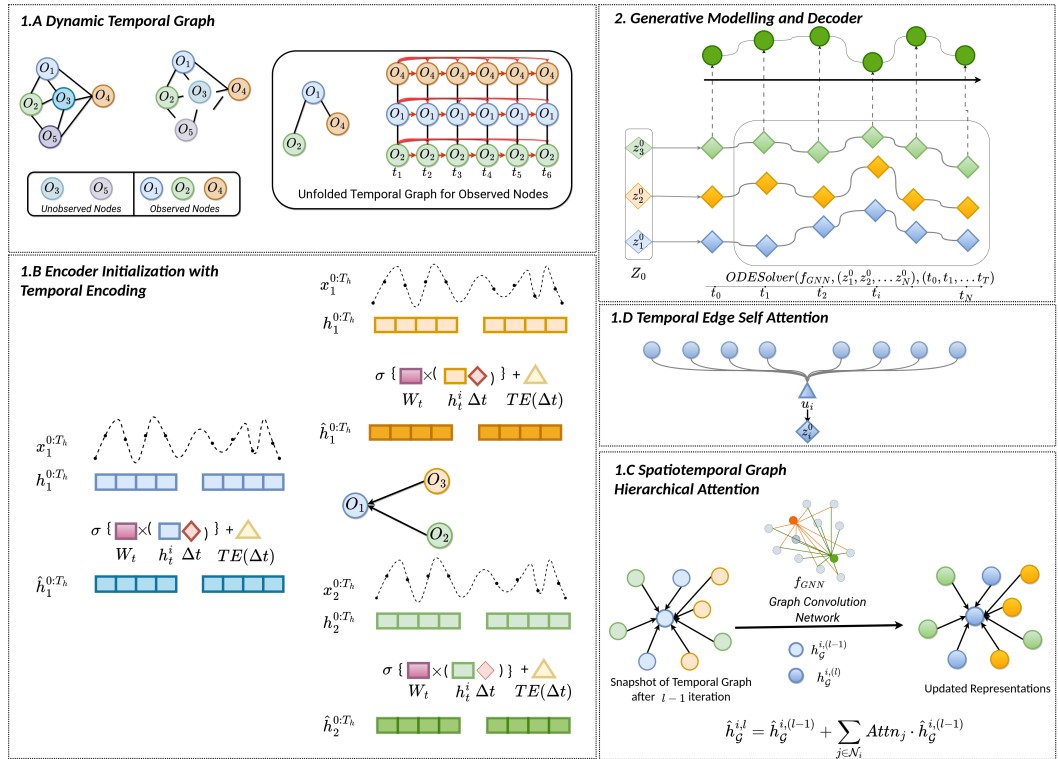

Figure 7: Design framework for encoder and decoder in STAGE Net (Best viewed in color.)

By generalizing to continuous scenarios, where $N_i$ denotes the set of immediate neighbors of object $o_i$, we reformulate it as:

$$\dot{z}_t^i := \frac{dz_t^i}{dt} = g_i(z_t^1, z_t^2, \ldots, z_t^N) = f_O\left(\sum_{j \in N_i} f_R([z_t^i, z_t^j])\right)$$

Here, the $||$ is the concatenation operations, and $f_O$, $f_R$ are two neural networks to capture the interaction of the latent system. The ODE function and the latent initial state $z_0^i$ will define the complete trajectories for each object. For the ode solver, we use the fourth-order Runge-Kutta method based on Chen et al. (2018) using the torchdiffeq python package (Chen (2018)).

## A.5 COMPUTATIONAL COMPLEXITY

In Figure 8, we present the computational complexity of our encoder's temporal graph. For evaluation, a spring system comprising 10 agents was simulated, generating simulations with varying distributions of visible and hidden agents. We observe the number of edges in the visible graph and temporal graph. For example, a model trained on data with 7 visible and 3 hidden agents yields an average of 10.5 edges for the visible agents. In contrast, our encoder's temporal graph, constructed over 30 timesteps, encompasses 13,048 edges. As the count of visible agents escalates, there's a corresponding increase in the temporal graph's edges, scaling at $\mathcal{O}((E + N)T^2)$, where $E$ and $N$ denote the edges and nodes of the initial interaction graph, excluding hidden agents. This relationship is illustrated in Figure 8a, which plots the average temporal edges against the average visible edges in the interaction graph. Additionally, Figure 8b showcases the GFLOPs of the Stage net's encoder in relation to the increment in visible agents.

| Hyperparameter | Value | Description |
|---|---|---|
| Scheduler | Cosine | Schedulerused to adjust the learning rate during training. |
| Test Data Size | 2000 | The number of samples in the test dataset. |
| Observation Std. Dev. | 0.01 | The standard deviation of the observation noise. |
| Number of Epochs | 100 | The number of times the learning algorithm will work through the entire training dataset. |
| Learning Rate | $5 \times 10^{-4}$ | The step size at each iteration while moving toward a minimum of the loss function. |
| Batch Size (Simulated) | 128 | The number of training examples utilized in one iteration. |
| Random Seed | 1991 | The seed used by the random number generator. |
| Dropout Rate | 0.2 | The probability of setting a neuron to zero during training. |
| Latent Size | 16 | The dimensionality of the latent space. |
| GNN Dimension | 64 | The dimensionality of the Graph Neural Network. |
| ODE Func Dimension | 128 | The dimensionality of the ODE Function. |
| GNN Layers | 2 | The number of layers in the Graph Neural Network. |
| Number of Heads in $z_0$ Encoder | 1 | The number of attention heads in the initial encoder. |
| ODE Func Layers | 1 | The number of layers in the ODE Function. |
| ODE Solver | RK4 | The method used to solve the Ordinary Differential Equation, Runge-Kutta of order 4 in this case. |
| Gradient Norm Clipping | 10 | The maximum allowed value for the gradient norm, used to prevent exploding gradients. |
| Number of Edge Types | 2 | The number of different types of edges in the graph. |
| L2 Regularization | $1 \times 10^{-3}$ | The weight decay parameter to prevent overfitting. |
| Optimizer | AdamW | The optimization algorithm used to minimize the loss function. |

Table 8: List and description of hyperparameters used in STAGE Net

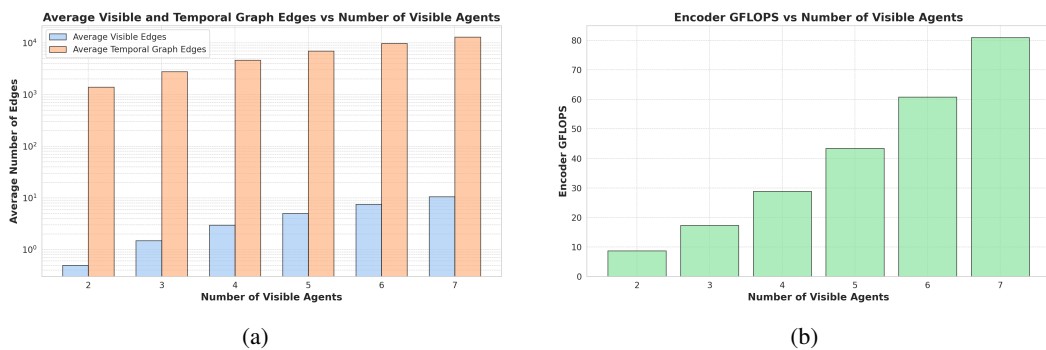

(a)  (b)

Figure 8: a.) A comparative representation of the average visible edges and average temporal graph edges against the number of visible agents. b.) Representation of Encoder GFLOPS against the Number of Visible Agents. Each bar signifies the computational complexity in GFLOPS of the encoder for the corresponding number of visible agents, highlighting the proportional increase in computational demand with the increase in visible agents

## B  ADDITIONAL EMPIRICAL RESULTS

### B.1  ANALYZING THE IMPACT OF HIDDEN AGENT INTERACTION STRENGTH ON MODEL PREDICTION

In this study, we study the influence of hidden agents by modifying the interaction strength amongst hidden agents in a spring system, with the interaction (coupling) strength systematically adjusted between 0.5 to 5.0. Concurrently, the interaction strength for visible agents is statically maintained at 1. For the spring dataset, interaction strength, symbolized as $F_{i,j}$, is quantified by the equation $F_{i,j} = -k(x_i - x_j)$, where $k$ represents the interaction strength between the entities $i$ and $j$.

The models were trained on a spring dataset with 50% observability consisting of 10,000 samples for each specified level of coupling and were subsequently evaluated on a separate test dataset, comprising 2,000 samples.

Figure 9 shows the $30^{th}$-step prediction error for all the baselines. A prominent observation from our experimental results is the exceptional and consistent performance of the StageNet model across all

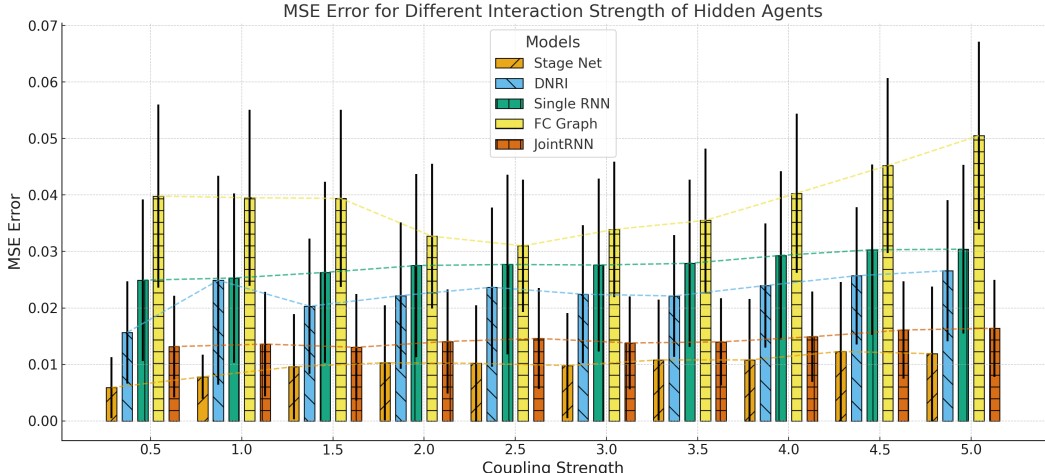

Figure 9: Mean Squared Error (MSE) values for the models as the interaction (coupling) strength is increased for the hidden agents.

degrees of coupling coefficients. StageNet not only exhibited a lower mean prediction error and low variance compared to the baseline models but also demonstrated remarkable stability, with its error rate not exhibiting a swift increase with the enhancement in interaction strength for hidden agents. This contrasts markedly with the other models, which showed a discernible upward error trend with increasing interaction strength. This empirical evidence underscores the resilience and dependability of StageNet in scenarios with varied interaction strengths, especially where the influence of hidden agents is pronounced in the system.

## B.2 DECIPHERING TEMPORAL CONTEXT FEATURE ATTENTION MAPS: THE INTERPLAY BETWEEN HIDDEN AGENTS, INFORMATION DENSITY, AND PREDICTION ACCURACY

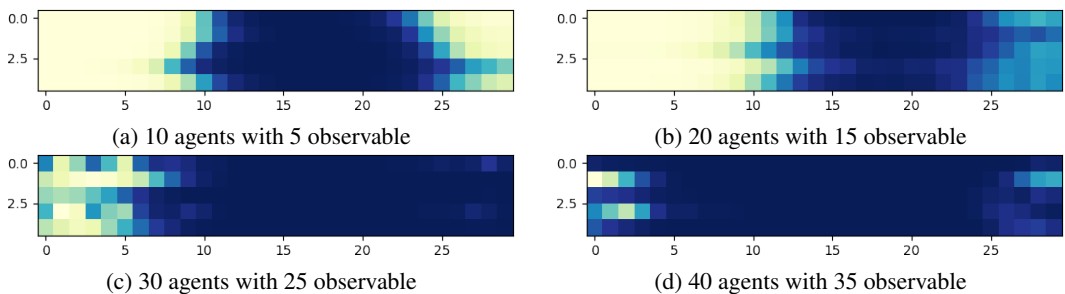

Figure 10: Temporal Context Feature Attention Maps: Visualization of temporal context feature attention across various configurations each with 50% observability

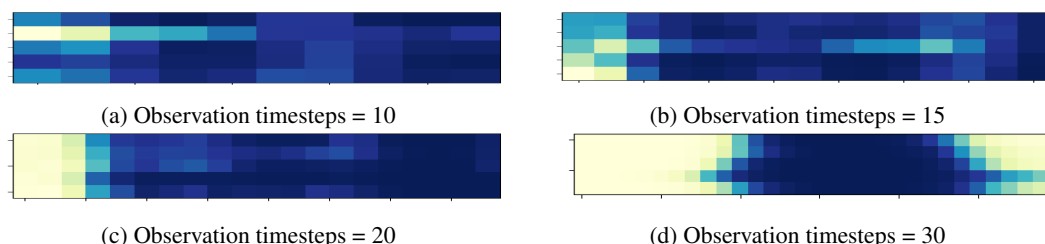

Figure 11: Temporal Context Feature Attention Maps: Feature attention maps applied to a system with 10 agents, including 50% unobservable agents, with variations in observation time.

Figure 10 visually illustrates temporal context feature attention maps for spring systems, each with a distinct proportion of hidden agents, ranging from 50% to 87.5%, while maintaining a constant count of five visible agents. The y-axis represents the index of the agent, and the x-axis plots the timesteps, with each row in the attention map representing the temporal attention values at different timesteps.

The attention values are scaled between 0 and 1, with yellow cells indicating a value of 0, and progressively darker shades of blue signifying attention values nearing 1. A critical observation is that as the proportion of hidden agents increases, the attention maps become densely populated with values of 1. This suggests that the network is utilizing every available timestep in the sequence to refine the precision of its future predictions. This transition to denser attention maps underscores a pivotal implication: the network, when faced with denser maps, is signaling a potential insufficiency in the available information. It is indicative of the network's increasing demand for more comprehensive data to optimize its predictive accuracy. Therefore, this density in attention values implies a heightened necessity to augment the number of timesteps observed. By extending the observation timesteps, we can cater to the network's increasing information needs, thereby enhancing the model's predictive accuracy and precision.

In essence, the densification of attention values in the maps is a clear indicator of the network's struggle with the available information, emphasizing the potential requirement to increase the observation timesteps to fulfill the network's information needs and, consequently, improve the accuracy of predictions.

Figure 11 provides further insight into this phenomenon by showcasing attention maps of four distinct models of a system, each consisting of 10 agents, 5 of which are hidden, across varied observation time periods, extending from 10 to 30 timesteps. A prominent observation from these maps is the progressive sparsification of the attention maps and a concurrent increase in predictive accuracy as the number of timesteps is increased. This is depicted in figure 12 where we plot the average MSE error for systems as their encoder's observation time is increased. This sparsification and enhanced accuracy suggest that the determination of an optimal observation period can be strategically made, contingent upon the number of hidden agents within the system. This analysis uncovers a crucial correlation: the higher the proportion of hidden agents in a system, the more extensive the observation period required to achieve accurate predictions. This denotes that systems with a greater number of hidden agents demand a more comprehensive observation framework to accurately capture the intricacies of the system dynamics and produce precise predictions.

In conclusion, the decrease in the density of attention maps and the corresponding enhancement in accuracy with extended timesteps emphasize the importance of selecting an optimal observation period, particularly in systems with a significant number of hidden agents. The insights derived from these attention maps serve as a valuable guide in the strategic selection of observation periods, facilitating the development of robust models capable of delivering precise predictions in a variety of scenarios.

### B.3 Performance of STAGE Net in Varied Topological Conditions with Fixed Number of visible Agents

In this experiment, we fix the quantity of visible agents within the system, while the number of hidden agents is subjected to variation. Figure 13 graphically represents the efficacy of the model, which has been trained on a spring system with 10 total agents out of which 5 are hidden agents. This is evaluated against systems with a diverse range of hidden agents, all the while maintaining the count of visible agents at 5.

The STAGE models consistently demonstrate superior performance over the baselines, regardless of the variations in the ratio of hidden to visible agents. This superiority of STAGE models is indicative of their robustness and adaptability across different scenarios, showcasing their ability to yield reliable results with different proportions of hidden and visible agents.

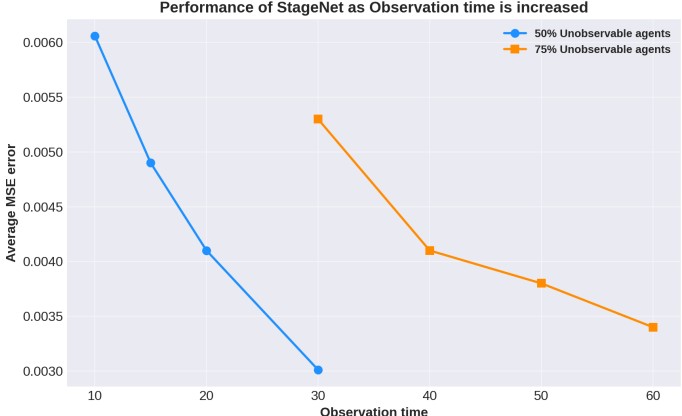

Figure 12: Prediction accuracy for two spring system with 50% and 75% unobservable agents as the observation time for encoder is increased

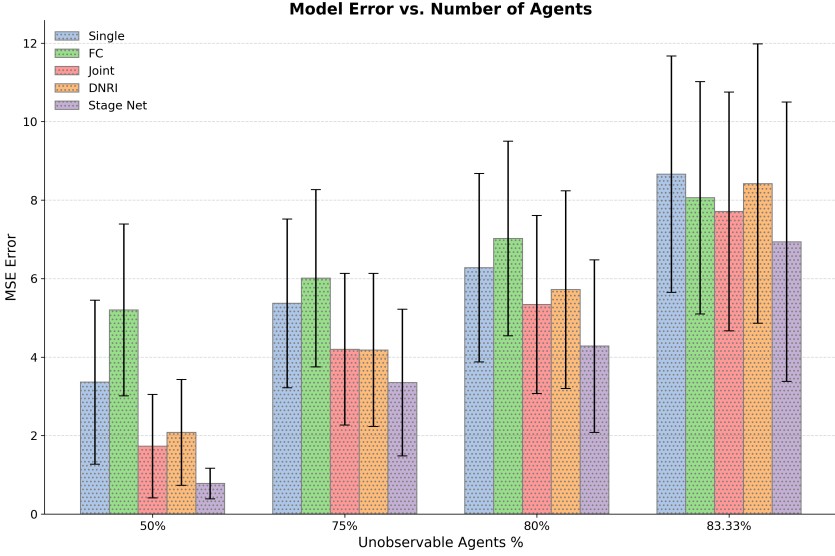

Figure 13: Mean Squared Error (MSE) values ($\times 10^{-2}$) for the model trained on a 10-5 configuration, while altering the total number of agents in the system, while keeping the visible agents fixed at 5.

### B.4 Model ablation: Impact of ODE Latent Dimension on Model Predictive Accuracy

For this experiment, we chose the spring system system with 50% unobservability and systematically varied the latent dimension of the ODE function. Our findings indicate that the optimal performance is achieved when the ODE latent size is set to 64, and performance deteriorates as the latent size deviates from this value. This phenomenon can be attributed to the following factors: When the latent size is kept small (e.g., 16 or 32), the model exhibits underfitting, meaning it struggles to capture the crucial characteristics and relationships within the multi-agent observations. Conversely, when the latent size is significantly increased (e.g., 512), it gives rise to the curse of dimensionality. In high-dimensional spaces, generalization becomes challenging as the model requires an extensive amount of data to effectively cover the feature space, leading to potentially poorer performance on the task at hand.

Table 9: Average MSE Error for different ODE latent dimension

| Size of ODE Latent | Average MSE Error |
| --- | --- |
| 16 | 0.0053 |
| 32 | 0.0041 |
| 64 | 0.0030 |
| 128 | 0.0037 |
| 256 | 0.0034 |
| 512 | 0.0041 |

### B.5 Evaluation of StageNet with Sensor Failures for Visible Agents

In this study, we address scenarios where observations for visible agents are intermittently unavailable due to random sensor failures. We consider two types of sensor failures: a) Asynchronous Sensor Failure, and b) Synchronous Sensor Failure. In the case of Synchronous Sensor Failure, all sensors for the visible agents fail simultaneously, leading to observations being available only at certain timesteps. Specifically, we randomly select 20 out of 30 timesteps, and the model receives observations only for these selected steps. Figure 14 illustrates the MSE error for a spring system with 10 agents, varying the percentage of unobservable agents. In contrast, during Asynchronous Sensor Failure, each agent's sensor fails independently, and we have observations for only 20 timesteps per agent. Figure 15 displays the MSE error for asynchronous sensor failure across the model. Compared to other models, StageNet demonstrates significantly lower error rates in both asynchronous and synchronous sensor failure scenarios.

### B.6 Robustness of StageNet against Noisy Data

This study further explores StageNet's resilience to noisy observations by training the model on noise-free data and evaluating it under Gaussian noise conditions (mean = 0) with varying standard deviations (0.001 to 0.1). In our investigation, we normalized the data before introducing noise to simulate real-world scenarios. We observed the model's performance in a spring system with 10 agents, particularly focusing on scenarios with different percentages of unobservable agents. Figure 16 depicts the Mean Squared Error (MSE) under Gaussian noise with a standard deviation of 0.1, highlighting StageNet's robustness even with high noise levels. Additionally, Figure 17 examines the MSE in a scenario where 50% of the agents are unobservable across different noise intensities, further illustrating the model's substantial resilience to noise. These results underscore StageNet's superior performance against noise, especially in comparison to baseline models.

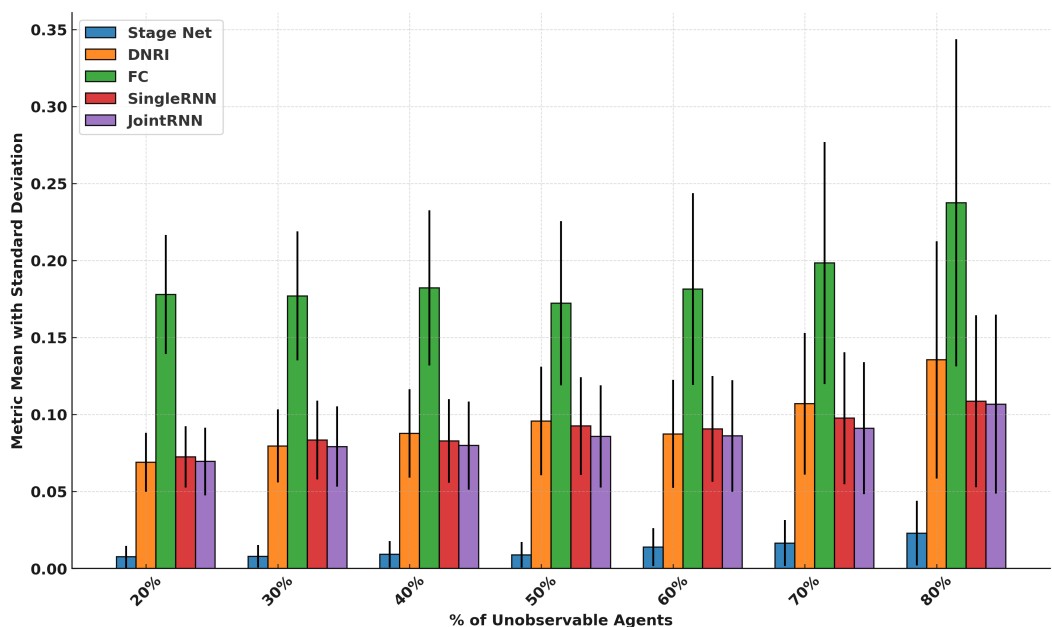

Figure 14: MSE error for synchronous sensor failure. Observations are randomly sampled for 20 steps out of 30 for all agents and provided to the model for evaluation.

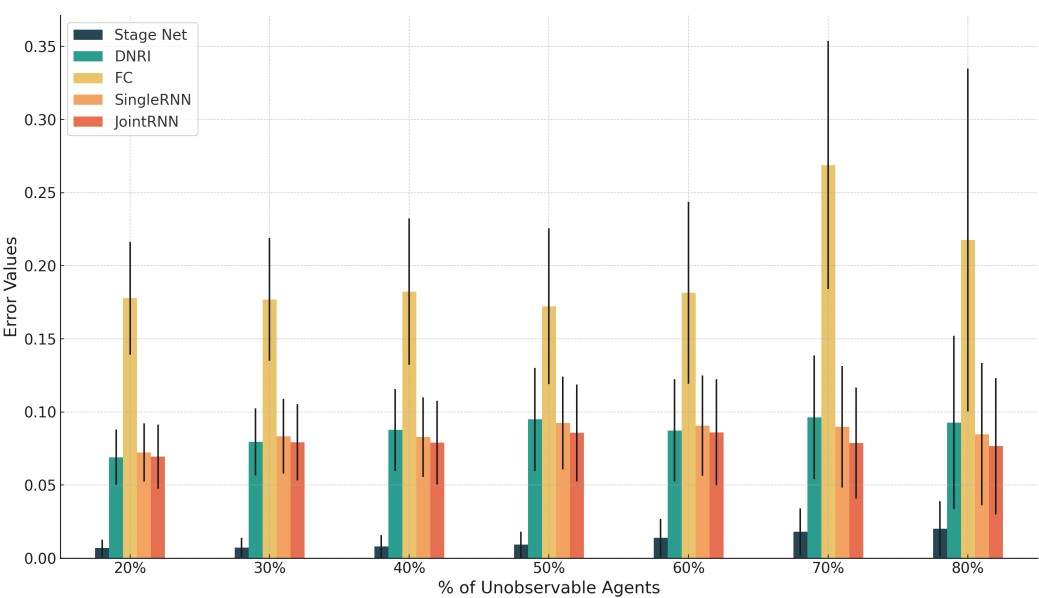

Figure 15: MSE error for asynchronous sensor failure. Observations are randomly sampled for 20 steps out of 30 for each agent and provided to the model for evaluation.

## B.7 EXPLORING SYSTEMS WITH HETEROGENEOUS AGENT CHARACTERISTICS

Our previous analysis primarily addressed systems with homogeneous agents, characterized by uniform dynamics across all entities. This section ventures into the realm of heterogeneous agents, introducing variability in agent dynamics. Specifically, we explore a spring system setup where each agent, as a heterogeneous entity, possesses distinct and unknown coupling parameters. In contrast to our earlier homogeneous agent experiments, which operated under a single coupling pa-

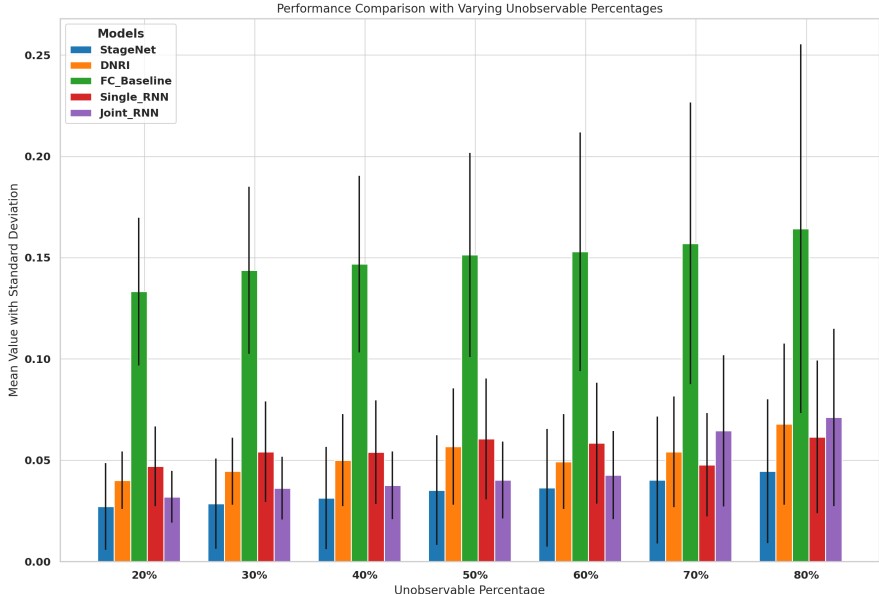

Figure 16: MSE Performance under Gaussian Noise (SD = 0.1) in a Spring System with 10 Agents, demonstrating StageNet's effective noise handling capabilities.

rameter setting for all agents, this study delves into varied configurations. We examine three distinct scenarios:

1. *Visible Heterogeneity, Hidden Homogeneity:* Only the visible agents exhibit heterogeneity, while hidden agents maintain homogeneous characteristics.

2. *Universal Heterogeneity:* Every agent in the system, both visible and hidden, is heterogeneous, with their coupling parameters randomly assigned.

3. *Hidden Heterogeneity, Visible Homogeneity:* This scenario reverses the first, with only hidden agents being heterogeneous.

Coupling Parameter Configurations: For the heterogeneous agents, we define three coupling parameter sets: a.) 3 types of agents: $\{0, 0.5, 1\}$, b.) 4 types of agents $\{0, 0.5, 1, 1.5\}$, and c.) 5 types of agents $\{0, 0.5, 1, 1.5, 2\}$.
During simulations, each heterogeneous agent's coupling parameter is randomly selected from these sets with uniform probability. Table 10 presents the error metrics for baseline models across different heterogeneous agent configurations, particularly when all agents are considered heterogeneous. We observe that baseline models struggle to capture the intricate dynamics of this setup, resulting in significantly higher error rates compared to our proposed model. Additional configurations and their outcomes are depicted in Figure 18, where similar trends are noted.

Table 10: Performance Metrics for Different Models for Heterogeneous Agents.

| Number of Het. Types | Stage Net Mean | Std | DNRI Mean | Std | FC Mean | Std | SingleRNN Mean | Std | JointRNN Mean | Std |
|---|---|---|---|---|---|---|---|---|---|---|
| 3 | 0.0104 | 0.0096 | 7.12 | 0.4076 | 2.28 | 0.39 | 2.92 | 0.26 | 3.55 | 0.31 |
| 4 | 0.0081 | 0.0077 | 7.16 | 0.38 | 2.26 | 0.377 | 2.91 | 0.2639 | 3.53 | 0.288 |
| 5 | 0.0089 | 0.0079 | 7.27 | 0.37 | 2.28 | 0.377 | 2.9 | 0.2454 | 3.5 | 0.27 |

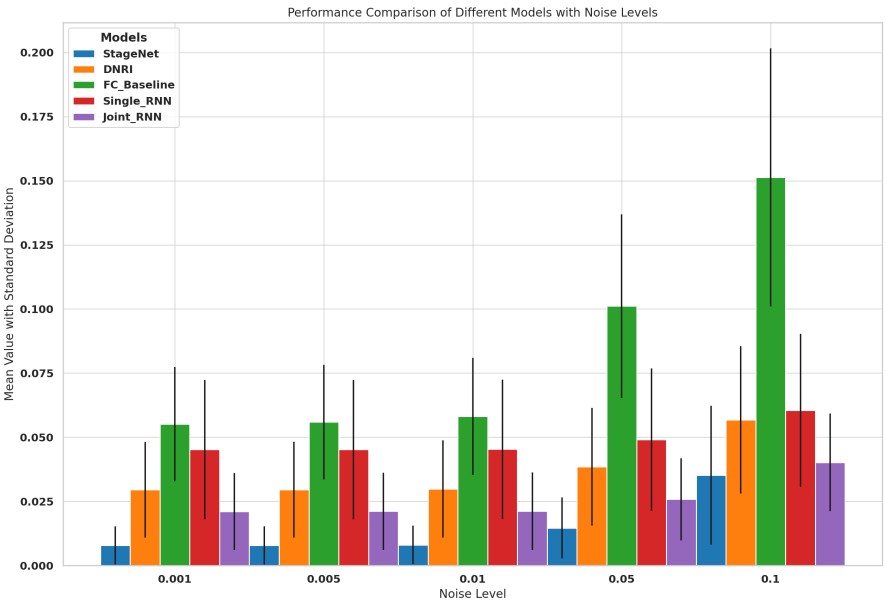

Figure 17: MSE Trends for a System with 50% Unobservable Agents across Various Noise Levels, showcasing the robustness of StageNet in complex, partially observable environments.

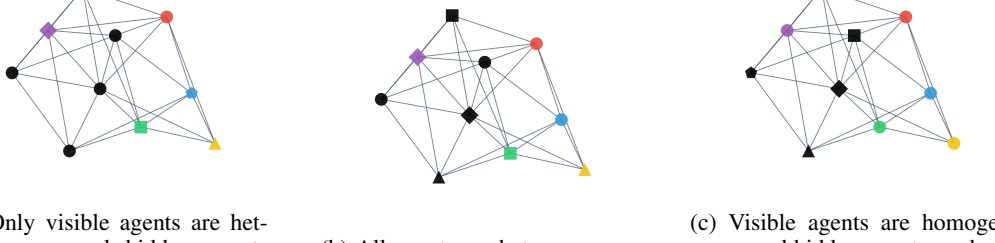

(a) Only visible agents are heterogeneous and hidden agents are homogeneous

(b) All agents are heterogeneous and randomly sampled

(c) Visible agents are homogeneous and hidden agents are heterogeneous

Figure 18: Different configurations of heterogeneous agents in our study

## C ANALYTICAL PROOFS

### C.1 DEFINITIONS

**Multisets and kernels for multisets** A *multiset* is a generalized notion of a set of a set, which accommodates multiple instances of its elements. We deliberate on multisets of features in $\mathbb{R}^d$, represented as:

$$\mathcal{X}^d = \left\{ \mathbf{x} \mid \mathbf{x} = \{\mathbf{x}_1, \ldots, \mathbf{x}_n\}, \text{ with each } \mathbf{x}_i \in \mathbb{R}^d \text{ for some } n \geq 1 \right\}$$

The cardinality of a multiset symbolized as $|\cdot|$, is determined by summing the multiplicities of its elements.

In this context, we assume the existence of a kernel on the space of multisets, represented as $K_{\mathrm{ms}} : \mathcal{X}^d \times \mathcal{X}^d \to \mathbb{R}$ and its either an exact or an approximate embedding, $\psi_{\mathrm{ms}} : \mathcal{X}^d \to \mathbb{R}^p$, such that

$$K_{\mathrm{ms}}(\mathbf{x}, \mathbf{x}') \approx \langle \psi_{\mathrm{ms}}(\mathbf{x}), \psi_{\mathrm{ms}}(\mathbf{x}') \rangle$$

**Temporal Graph**  Let $G(V(t), E(t))$ be the graph with nodes $V(t)$ and edges $E(t)$ at time $t$. Let $G'$ be a subgraph of $G$ with observed nodes $x_1(t), x_2(t), \ldots, x_N(t)$. The *Temporal Graph* $T'$ can be defined as a multiset of the states of graph $G'$ at different time points, represented as: $T' = \{G'(t_1), G'(t_2), \ldots, G'(t_r)\}$ where each $G'(t_i)$ is a member of the multiset representing the state of graph $G'$ at time $t_i$, and additional temporal edges are added between nodes in $G'(t_i)$ and $G'(t_{i+1})$ for all $i = 1, 2, \ldots, r-1$ to represent the temporal connections between the different states of a graph $G'$.

In the derivation of all our analytical results, we base our arguments on the subsequent assumptions:

**Assumptions:**

1. We assume the embedding of each individual node, $x_i(t)$, to conform to a multivariate Gaussian distribution, parametrized by $\theta = \{\mu, \Sigma\}$.

2. The embedding of the multiset, $X_i$, is hypothesized to adhere to a Gaussian Mixture Model (GMM) with $K$ components, described by parameters $\phi = \{\pi, \mu, \Sigma\}$. Here, $\pi$ signifies the mixture weights, $\mu$ represents the means, and $\Sigma$ defines the covariance matrices of the components.

3. $I(\theta; N)$ represent the Fisher Information Matrix (FIM) as a function of the parameter $\theta$ and the number of observed nodes $N$.

4. The Fisher Information is a differentiable function with respect to the number of observed nodes.

## C.2 Analysis of Fisher Information in Multiset Embeddings for Temporal Graph

**Theorem 1** *The Fisher information of the embedding of the multiset $X_i$ is greater than the Fisher information of the embedding of each individual element $x_i(t)$ i.e., $det(J(\phi) > det(I(\theta))$*

**Proof:** Let the probability density function representing the embedding of node $x_i$ at time $t$ be $f(x_i(t); \theta)$, parameterized by $\theta$. Similarly, let the probability density function representing the embedding of the multiset $X_i$ be $g(X_i; \phi)$, parameterized by $\phi$. Each individual node embedding $x_i(t)$ is assumed to follow a Gaussian distribution:

$$f(x_i(t); \mu, \sigma^2) = \frac{1}{\sqrt{2\pi\sigma^2}} \exp\left(-\frac{(x_i(t) - \mu)^2}{2\sigma^2}\right) \tag{9}$$

The multiset embedding $X_i$ is assumed to follow a Gaussian Mixture Model with $K$ components:

$$g(X_i; \pi, \mu, \Sigma) = \sum_{k=1}^{K} \pi_k \mathcal{N}(X_i; \mu_k, \Sigma_k) \tag{10}$$

If the Fisher information for an individual node is given by $T(\theta)$ and the Fisher information of the multiset $X_i$ is given as $J(\phi)$, then:

$$I(\theta) = \mathbb{E}\left[\left(\frac{d}{d\theta} \log f(x_i(t); \theta)\right)^2\right] \Rightarrow J(\phi) = \mathbb{E}\left[\left(\frac{d}{d\phi} \log g(X_i; \phi)\right)^2\right] \tag{11}$$

Let's assume that the covariate distribution between any two nodes $x_i(t)$ and $x_j(t)$ is Gaussian, with parameters $\theta = \{\mu_{ij}, \sigma_{ij}^2\}$, where $\mu_{ij}$ is mean and $\sigma_{ij}^2$ is the variance of the Gaussian distribution representing the covariate between nodes $i$ and $j$. Given the Gaussian covariate distribution between the nodes, the Fisher Information for the covariate distribution between nodes $i$ and $j$ is given by:

$$I(\theta) = \begin{bmatrix} \frac{1}{\sigma_{ij}^2} & Cov(\mu, \sigma^2) \\ Cov(\mu, \sigma^2) & \frac{1}{2\sigma_{ij}^4} \end{bmatrix}$$

For a Gaussian Mixture Model, the Fisher information matrix $J(\phi)$ where $\phi = \{\pi, \mu, \Sigma\}$ depends on the derivatives of the log-likelihood with respect to the parameters. The elements of the Fisher information matrix are given by the expected second derivatives of the log-likelihood, which can be computed using the Expectation-Maximization (EM) algorithm. Assume that the embedding of node $x_i(t)$ follows a Gaussian distribution with mean $\mu$ and variance $\sigma^2$, both parameterized by $\theta$. The Fisher information, $I(\theta)$, for this node is derived as follows:

$$I(\theta) = \mathbb{E}\left[\left(\frac{d}{d\theta} \log f(x_i(t); \mu, \sigma^2)\right)^2\right] \tag{12}$$

Assume that the embedding of the multiset $X_i$ follows a Gaussian mixture model with $K$ components, each with its own mean $\mu_k$ and variance $\sigma_k^2$, all parameterized by $\phi$. The Fisher information, $J(\phi)$, for this multiset is derived as follows:

$$J(\phi) = \mathbb{E}\left[\left(\frac{d}{d\phi} \log g(X_i; \{\mu_k, \sigma_k^2\}_{k=1}^K)\right)^2\right] \tag{13}$$

To compare $J(\phi)$ and $I(\theta)$, we need to compare the respective Fisher information matrices.

Since these matrices are of different dimensions, a direct comparison is not straightforward. However, we can compare the determinant of the Fisher information matrices as a scalar representation of the information contained in the embeddings. We aim to compare the determinant of the Fisher Information Matrix for a Gaussian Mixture Model (GMM) with that of a Gaussian distribution. We will symbolically represent the Fisher Information Matrix for a GMM and derive its determinant to compare with the determinant of the Fisher Information Matrix for a Gaussian distribution. Let's consider a GMM with $K$ components, each with parameters $\phi_k = \{\pi_k, \mu_k, \Sigma_k\}$, where $\pi_k$ is the weight, $\mu_k$ is the mean, and $\Sigma_k$ is the covariance matrix of the $k$-th component. The log-likelihood for the GMM is given by:

$$\log L(\phi) = \sum_{i=1}^N \log\left(\sum_{k=1}^K \pi_k \mathcal{N}(x_i; \mu_k, \Sigma_k)\right) \tag{14}$$

The Fisher Information Matrix, $J(\phi)$, for the GMM is a block-diagonal matrix, where each block corresponds to the Fisher Information Matrix for the parameters of component $k$, $J(\phi_k)$. Each block, $J(\phi_k)$, can be represented symbolically as:

$$[J(\phi_k)]_{mn} = \mathbb{E}\left[\frac{\partial^2 \log L(\phi)}{\partial \phi_{km} \partial \phi_{kn}}\right] \tag{15}$$

Now, considering Gaussian covariance between the components, we need to consider the interaction between the components of the Gaussian Mixture Model (GMM) and derive the Fisher Information Matrix accordingly. When the components are not independent, the blocks of the Fisher Information Matrix are not necessarily diagonal, and the off-diagonal elements represent the covariance between the components. Let's denote the covariance between component $k$ and component $l$ as $\Sigma_{kl}$. The Fisher Information Matrix, $J(\phi)$, for the GMM with covariance can be represented as:

$$[J(\phi)]_{mn} = \mathbb{E}\left[\frac{\partial^2 \log L(\phi)}{\partial \phi_{km} \partial \phi_{ln}}\right] + \Sigma_{kl}$$

The Fisher Information for the GMM can be expressed as a weighted sum of the Fisher Information of the individual components:

$$J_{X_i}(\phi) = \sum_{k=1}^K \pi_k I_{x_i}(\theta_k; N_k)$$

where $\pi_k$ are the mixture weights, $\theta_k$ are the parameters for each component, and $N_k$ is the number of observations assigned to the k-th component.

Let $J(\phi)$ denote the Fisher Information Matrix with covariance, represented as a block matrix:

$$J(\phi) = \begin{bmatrix} J(\phi_k) & \Sigma_{kl} \\ \Sigma_{lk} & J(\phi_l) \end{bmatrix}$$

where $J(\phi_k)$ and $J(\phi_l)$ are the Fisher Information Matrices for individual components, and $\Sigma_{kl}$ and $\Sigma_{lk}$ are the covariance matrices between the components.

There can be two cases that arise here.

**Case I:** $\Sigma_{kl} = \Sigma_{lk} = 0$ Then, the determinant of $J(\phi)$ is strictly greater than the product of the determinants of the individual Fisher Information Matrices, i.e.,

$$\det(J(\phi)) > \det(J(\phi_k)) \cdot \det(J(\phi_l))$$

i.e the determinant of the Fisher Information Matrix with covariance for two components is greater than the determinant of the Fisher Information Matrix when they are independent. Given that the determinant of the Fisher Information Matrix for the GMM with covariance is greater than the determinant of the Fisher Information Matrix, it is evident that the multiset embedding $X_i$ will contain more information about the parameters than the individual node embedding $x_i(t)$ when considering Gaussian covariance between the components. Thus, since the determinant of $J(\phi)$ is greater than the determinant of $I(\theta)$, then it can be concluded that the multiset embedding contains more information about the parameters than the individual node embedding.

**Case II:** $\Sigma_{kl}, \Sigma_{lk} \neq 0$

When there is covariance between two Gaussian components in a GMM, the elements in $J(\phi)$ representing the covariance between these components would be non-zero, symbolizing the interaction between the components. To prove the inequality $|J(\phi)| > |I(\theta)|$, let us elaborate that the determinant of the Fisher Information Matrix, $|J(\phi)|$, for the GMM with covariance, will typically be greater due to the additional terms representing the interaction between the components along with the individual components' information. Let us assume there are $K$ Gaussian components in the GMM, each with its mean and variance, and let's denote the covariance between the i-th and j-th components as $cov(i, j)$. The determinant of $J(\phi)$ would be the sum of the determinants of the individual components plus the terms representing the covariance interaction between the components:

$$|J(\phi)| \approx \sum_{i=1}^{K} |I_i| + \sum_{i \neq j} cov(i, j)$$

Since the covariance terms represent additional information not present in a single Gaussian component, it would generally contribute to a greater determinant of $J(\phi)$ as compared to $|I(\theta)|$:

$$|J(\phi)| > |I(\theta)|$$

Hence, for both cases, we proved that the Fisher information of the embedding of the multiset $X_i$ is greater than the Fisher information of the embedding of each individual element $x_i(t)$.

**Theorem 2** *Given the reduced temporal graph $T'$, the corresponding reduced spatial graph $G'$, and the static spatial graph $G$, if the Fisher information of the embedding of $T'$ exceeds the Fisher information of the embedding of $G'$, i.e.,*

$$I(T') > I(G')$$

*then it follows that the covariance of the reduced temporal graph, $Cov(T')$, is less than the covariance of the reduced spatial graph, $Cov(G')$, represented as:*

$$Cov(T') < Cov(G')$$

**Proof:** Let $I(T')$ and $I(G')$ denote the Fisher Information in the reduced temporal graph $T'$ and the reduced spatial graph $G'$ respectively, both of which are derived from a complete graph $G$. The Fisher Information Matrix for each graph is computed based on the observed nodes and their relationships within the respective graphs.

From definition, the temporal graph $T'$ is the multiset representation of a sequence of spatial graphs $G'$ at different time points. According to Cramér–Rao Lower Bound (CRLB), i

$X = (X_1, X_2, \ldots, X_n)$ be a random vector with probability density function $f(\mathbf{x}; \boldsymbol{\theta})$, where $\boldsymbol{\theta} = (\theta_1, \theta_2, \ldots, \theta_k)$ is a vector of parameters of interest. Let $\mathbf{T}(\mathbf{X}) = (T_1(\mathbf{X}), T_2(\mathbf{X}), \ldots, T_k(\mathbf{X}))$ be an unbiased estimator of $\boldsymbol{\theta}$, i.e., $\mathbb{E}[\mathbf{T}(\mathbf{X})] = \boldsymbol{\theta}$. Then, for any unbiased estimator $\mathbf{T}(\mathbf{X})$, the covariance matrix of $\mathbf{T}(\mathbf{X})$ satisfies:

$$\text{Cov}(\mathbf{T}(\mathbf{X}), \boldsymbol{\theta}) \geq \mathbf{I}(\boldsymbol{\theta})^{-1},$$

where $\mathbf{I}(\boldsymbol{\theta})$ is the Fisher Information matrix of the random vector $\mathbf{X}$ with respect to the parameter vector $\boldsymbol{\theta}$ Thus, we can conclude that the Fisher information of the embedding of the $T'$ is greater than the Fisher information on the embedding of each spatial graph $G'$ at any timestep.

$$I(T') > I(G')$$

Thus, it is concluded that based on the construction and inherent properties of the temporal graph $T'$ and the spatial graph $G'$, the reduced temporal graph $T'$ retains more information than the reduced spatial graph $G'$.

The Fisher information of the embedding of the $T'$ is greater than the Fisher information of the embedding of $G'$:

$$I(T') > I(G')$$

Consequently, due to the inverse relationship between Fisher Information and covariance:

$$I(T')^{-1} < I(G')^{-1}$$

Applying the Cramér-Rao Bound, we relate the inverses of the Fisher Information to the covariances of the estimators:

$$\text{Cov}(T') \leq I(T')^{-1} < I(G')^{-1} \leq \text{Cov}(G')$$
$$\Rightarrow \text{Cov}(T') < \text{Cov}(G')$$

Thus, it is concluded that the covariance of the reduced temporal graph $T'$ serves as a more accurate estimator for the complete graph $G$ compared to the covariance of the reduced spatial graph $G'$.

## D  BROADER IMPACT

Many often we do not operate in complete information settings for these complex co-evolving systems and addressing the practical challenges of measuring the entire system, our work provides valuable insights into the analysis of subgraphs in various domains. This has implications for fields such as protein-protein interactions, metabolic networks, planetary systems, and robotic systems, where complete agent measurements are often unattainable. Additionally, our framework is beneficial for large-scale networks that are either computationally intensive to handle, as it enables deliberate sampling of smaller subnetworks for analysis or have sensor failures thereby having incomplete knowledge of the system's degrees of freedom. This has practical implications for researchers and practitioners working with complex networks, allowing them to focus their analysis on representative subgraphs while maintaining reasonable accuracy.

## E  ADDITIONAL VISUALIZATIONS

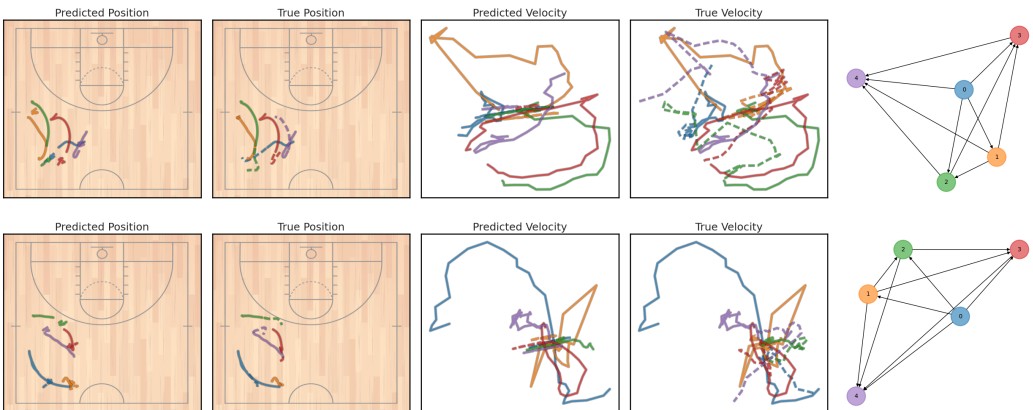

Figure 19: Visualizations depicting predictive trajectories for basketball dataset involving 5 players. Dotted lines represent predicted trajectories, while solid lines represent observed trajectories.

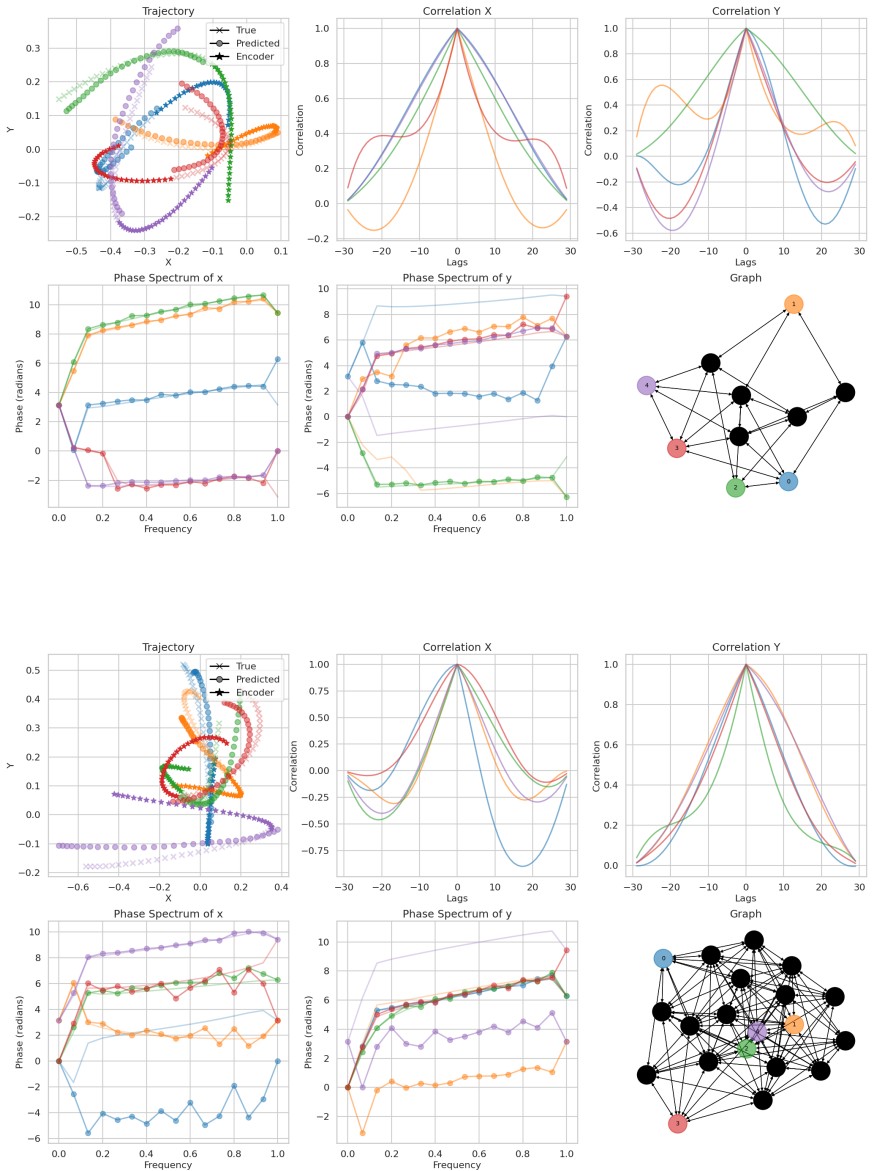

Figure 20: Visualizations depicting predictive trajectories for spring systems involving varying degrees of hidden agents. In the top row, a system with 10 agents and 50% hidden agents is shown, while the bottom row displays a system with 20 agents and 75% hidden agents. We also plot correlation and phase plots for both the systems as correlation plots for variables X and Y across different lags help in determining the time lag between predictions and observations, enabling a better understanding of the temporal dynamics in the data.

