# OpenReview forum: "STAGE Net: Spatio-Temporal Attention-based Graph Encoding for Learning Multi-Agent Interactions in the presence of Hidden Agents"
_ICLR.cc/2024/Conference — ICLR 2024 Conference Withdrawn Submission_

### Official Review · Reviewer_d9Cc · 2023-10-28

**Soundness:** 3 good
**Presentation:** 3 good
**Contribution:** 2 fair
**Rating:** 5
**Confidence:** 4

**Summary:**

This paper proposed a Spatio-Temporal Graph Attention Network (called STAGE Net) to learn multi-agent dynamics where some agents are completely unobserved (hidden) all the time. The network used the spatiotemporal attention mechanism with neural inter-node messaging to capture high-level behavioral semantics of the multi-agent system. They showed analytical results motivating STAGE Net using spatiotemporal graphs with time anchors to effectively model complex multi-agent interactions with unobserved agents and no prior information about interaction graph topology. They also show the evaluation results on multi-agent simulations with spring and charged dynamics and a motion trajectory dataset. STAGE Net outperformed existing multiagent interaction modeling networks in predicting trajectories of complex multiagent interactions even when having a large number of unobserved agents.

**Strengths:**

- The paper developed a framework to address the problem about complex multi-agent systems with unobserved agents.
- The STAGE Net used a dynamic spatiotemporal graph to model structural information across time using observations from visible nodes to recover knowledge representations missing due to unobserved agents.
- They performed theoretical analyses provided on why the spatio-temporal graph obtained superior representations compared to just using the visible agents' interaction graph.
- The experimental results showed that the method outperformed several baselines on multiple datasets with spring, charged, and motion trajectory dynamics.

**Weaknesses:**

- There have been many spatiotemporal graph attention networks in previous work (in Google scholar, 56 items), but the proposed method’s name is based on this. Can the authors reconsider the name and clarify the differences from these papers? In other words, the novelty of the methodology in STAGE Net was unclear and in the experiments, some similar networks can be compared (the baselines were old; dNRI was proposed in 2020).
- In the experiments, the model performances were evaluated extensively on simulated physics datasets and single-agent (and multi-joint with physical constraints) CMU dataset, but real-world multi-agent trajectory datasets can be used to demonstrate applicability.
- As written in conclusion, there is no analysis provided on how the performance changes for heterogeneous agents with diverse dynamics, but this may not be a fatal problem in this paper (considered as the limitation).

**Questions:**

- Again, there have been many spatiotemporal graph attention networks in previous work. Can the authors reconsider the name and clarify the differences from these papers? In other words, the novelty of the methodology in STAGE Net was unclear.
- P3: may be a typo:  “the is”
- P4: subscripts of \mathcal{X} (time interval) in the third and fourth lines of the first paragraph in 2.2.2. Did they correspond with the definition of 2.2.1 and are they correct? Can the t be arbitrary and is the T_h necessary for the former?
- P4: The definitions of the nonlinear activation function and the concatenation operation after Eq. (2) can be moved to Eq. (1).
- Experiments (methods): again, some similar (spatiotemporal graph attention) networks can be compared (the baselines were old; dNRI was proposed in 2020).
- Experiments (datasets): again, the model performances were evaluated extensively on simulated physics datasets and single-agent (and multi-joint with physical constraints) CMU dataset, but real-world multi-agent trajectory datasets can be used to demonstrate applicability. For example, dNRI paper used an NBA basketball dataset.

---

> ### Author Response · Authors · 2023-11-23
> **Heterogenous Agents**
>
> > As written in conclusion, there is no analysis provided on how the performance changes for heterogeneous agents with diverse dynamics, but this may not be a fatal problem in this paper (considered as the limitation).
>
>
> We thank the reviewer for highlighting the importance of analyzing the performance of our model in scenarios involving heterogeneous agents with diverse dynamics. In response to this valuable feedback, we have included a new section in our manuscript titled "Exploring Systems with Heterogeneous Agent Characteristics." This section specifically addresses the dynamics of systems comprising agents with varying characteristics, a crucial aspect that was not fully explored in our initial submission.
>
> In this new section, we investigate the implications of heterogeneity in agent dynamics through a series of experiments set in a spring system. Here, each agent is treated as a heterogeneous entity with distinct and unknown coupling parameters. We explore three distinct scenarios to comprehensively understand the impact of agent heterogeneity:
>
> 1. **Visible Heterogeneity, Hidden Homogeneity**: In this scenario, only the visible agents exhibit heterogeneity in their dynamics, while the hidden agents maintain homogeneous characteristics.
>
> 2. **Universal Heterogeneity**: This scenario considers every agent in the system, both visible and hidden, as heterogeneous. Each agent's coupling parameters are randomly assigned, introducing a high degree of variability in the system.
>
> 3. **Hidden Heterogeneity, Visible Homogeneity**: This setup is the inverse of the first scenario, where only the hidden agents are heterogeneous, while the visible agents are homogeneous.
>
> For each of these scenarios, we define three sets of coupling parameters to represent different levels of heterogeneity among agents. During our simulations, each heterogeneous agent's coupling parameter is randomly selected from these sets, introducing a controlled yet significant level of complexity to the system dynamics.
>
> Our empirical results for prediction error, presented in Table below demonstrate that baseline models struggle to accurately capture the intricate dynamics introduced by agent heterogeneity. In contrast, our proposed model shows a significantly better performance in handling these complex scenarios, as evidenced by lower error rates and more accurate predictions.
>
> *Performance MSE Metrics for Different Models for Heterogeneous Agents*
> | **Number of Het. Types** | **Stage Net**       | **DNRI**           | **FC**            | **SingleRNN**         | **JointRNN**        |
> |:------------------------:|:-------------------:|:------------------:|:-----------------:|:---------------------:|:-------------------:|
> | 3                        | 0.0104 ± 0.0096     | 7.12 ± 0.4076      | 2.28 ± 0.39       | 2.92 ± 0.26           | 3.55 ± 0.31         |
> | 4                        | 0.0081 ± 0.0077     | 7.16 ± 0.38        | 2.26 ± 0.377      | 2.91 ± 0.2639         | 3.53 ± 0.288        |
> | 5                        | 0.0089 ± 0.0079     | 7.27 ± 0.37        | 2.28 ± 0.377      | 2.9 ± 0.2454          | 3.5 ± 0.27          |
>
>
> By incorporating this analysis into our manuscript, we aim to address the previously identified limitation and provide a more comprehensive understanding of our model's capabilities in diverse and complex multi-agent systems.
> **Refer to Section B.7 of Supp for more details and plots**

---

> ### Author Response · Authors · 2023-11-23
> **Real World Datasets**
>
> > In the experiments, the model performances were evaluated extensively on simulated physics datasets and single-agent (and multi-joint with physical constraints) CMU dataset, but real-world multi-agent trajectory datasets can be used to demonstrate applicability.
>
> Thank you for your comment. We have addressed this concern by incorporating two real-world datasets to showcase the applicability of our approach:
>
> a. **CMU Motion Mocap dataset**: We leveraged the Carnegie Mellon University (CMU) Motion Capture dataset, a comprehensive collection of human motion capture data. This dataset encompasses a wide range of activities, including walking, running, and dancing, recorded from diverse subjects. Specifically, we focused on Subject 35 and their walking trajectories, comprising 8,063 frames, each documenting 31 specific joints. Each joint is treated as an agent connected to others through linkages. The dataset attributes, encompassing position and velocity, were normalized to have a maximum absolute value of 1. Our models were trained on 30-timestep sequences and evaluated on sequences of the same length. The table below presents prediction results for motion datasets with different sets of joints randomly hidden during training. Our network consistently outperforms baseline models in this context.
>
> | Unobserved Joints | 0    | 5    | 10   | 15   | 20   |
> |-------------------|------|------|------|------|------|
> | MultiBlock RNN    | 0.17 | 0.18 | 0.16 | 0.18 | 0.13 |
> | FC Graph          | 0.14 | 0.13 | 0.19 | 0.16 | 0.16 |
> | JointRNN          | 0.30 | 0.34 | 0.28 | 0.23 | 0.23 |
> | D-NRI             | 0.16 | 0.30 | 0.17 | 0.30 | 0.20 |
> | STAGE Net (ours)  | 0.11 | 0.13 | 0.11 | 0.10 | 0.13 |
>
>
> b. **Basketball Dataset**: Our second real-world dataset involves detailed 2D position and velocity trajectories of the offensive team, consisting of 5 players. Trajectories were initially divided into 49 frames, capturing approximately 8 seconds of gameplay. During training, models were trained on the first 30 frames from the training trajectories. For evaluation, models were presented with input data comprising sampled trajectories from the initial 30 frames, with sampling adjusted based on temporal sparsity. For example, with 10% temporal sparsity, we selected 27 observations from the initial 30 for each player, and the models predicted the subsequent 19 frames. The table below displays the results of the basketball dataset, where only 50% of the agents are observable. Introducing temporal sparsity through random sparse sampling of encoder observations, our observations indicate that STAGE Net outperforms baseline models in scenarios involving concealed agents and limited temporal observability.
>
> | \% Temporal Observability | STAGE Net | DNRI | FC  | Joint RNN | Single RNN |
> |---------------------------|-----------|------|-----|-----------|------------|
> | 33%                       | 1.40 ± 2.45 | 2.60 ± 1.79 | 2.67 ± 1.67 | 1.94 ± 1.51 | 1.86 ± 1.47 |
> | 50%                       | 1.36 ± 2.39 | 2.35 ± 1.57 | 2.45 ± 1.90 | 1.62 ± 1.29 | 1.55 ± 1.26 |
> | 66%                       | 1.37 ± 2.40 | 2.00 ± 1.33 | 2.21 ± 1.25 | 1.29 ± 1.03 | 1.22 ± 1.00 |
> | 100%                      | 1.32 ± 2.33 | 2.069 ± 1.23 | 1.72 ± 0.93 | 1.022 ± 7.6 | 0.98 ± 0.77 |
> These real-world datasets serve to validate the effectiveness of our approach in practical scenarios.

---

> ### Author Response · Authors · 2023-11-23
> **Spatiotemporal graph attention networks**
>
> > Again, there have been many spatiotemporal graph attention networks in previous work. Can the authors reconsider the name and clarify the differences from these papers? In other words, the novelty of the methodology in STAGE Net was unclear.
>
> We thank the reviewer for their insightful comments. Our response aims to clarify the distinctiveness of our methodology and its novelty in comparison to existing works:
>
> 1.  **Problem Statement**: Our study uniquely addresses a scenario characterized by inherently unobservable agents within multi-agent systems, a challenge not commonly tackled in existing research. Most prior studies assume complete observability of agents, either with sparse or continuous sampling. Our work explores a more complex scenario where certain agents remain perpetually unobservable. This specificity sets our research apart, as it deals with a system having fewer independent degrees of freedom than its intrinsic dimension.
>
>
>    | **Scenario** | **Description of Problem** | **References** |
>    |--------------|----------------------------|----------------|
> | Complete observability with known interaction topology | Multi-agent systems where all agents are observable at all times, with a known interaction topology, facilitating the modeling process. | Watters et al., 2017 |
> | Complete observability with unknown interaction topology | All agents are observable at all times; however, the interaction topology is not predefined and must be inferred from observational data. | Alahi et al., 2016; Banijamali, 2022; Graber et al; Kips et al. 2018; Alet et al. 2019; Steenkiste et al 2018, Santoro et. al. 2017 |
> | Complete observability with Irregular sampling of observations or temporally sparse data | All agents are observable but the observation events are sporadic or irregular, leading to temporal data sparsity. | Rubanova et al., 2019; Zhu et al., 2021; Huang et al. 2020; Marisca et al., 2022; Sun et al., 2019 |
> | Only few agents observable with sparse temporal sampling and unknown interaction topology | Not all agents are observable, with some never being observed, coupled with sparse temporal data collection. | (Ours) |
>
>
> 2. **Spatio-Temporal Graph**: While it is true that methods like the Dynamics Neural Relational Inference (dNRI) also utilize spatio-temporal graphs, our approach distinctly integrates the temporal dimension within the spatial domain. In our method, we construct a graph encompassing both spatial and temporal edges for each timestep. Each agent's observation is represented as a temporal node, with temporal relations established among agents. These nodes feature unique vectors encapsulating spatial location and velocity, further enriched by time anchors that reflect the chronological context of each observation. Our edge construction process is based on the temporal disparity between nodes, with edges formulated within a predefined temporal threshold. This approach leads to the formation of a comprehensive spatiotemporal graph, denoted as $\mathcal{G}$, on which we apply a graph neural network. This network is instrumental in approximating the initial latent posterior distribution, effectively facilitating long-term predictions in partially observable systems. This methodology differs significantly from the typical approach of processing spatial information through an attention network (like in papers cited below) and then applying temporal modeling via LSTM or similar frameworks. Our integrated spatio-temporal representation enables a more nuanced understanding of agent interactions over time, setting our method apart from existing techniques. In conclusion, our paper not only addresses a less-explored problem in the realm of multi-agent systems but also introduces a novel approach in constructing and utilizing spatiotemporal graphs. These aspects underscore the originality and contribution of our work to the field.
>
> *[Spatial and Temporal Attention Networks]*
>
> a.  Spatial-Temporal Graph Attention Networks: A Deep Learning Approach for Traffic Forecasting
>
> b. ST-GRAT: A Novel Spatio-temporal Graph Attention Network for Accurately Forecasting Dynamically Changing Road Speed
>
> c. Dynamic Neural Relational Inference

---

> ### Author Response · Authors · 2023-11-23
> **Typos and other formatting changes**
>
> > P3: may be a typo: “the is”
>
> Thank you for pointing out this potential typo. We have carefully reviewed the mentioned section and corrected the typo. Ensuring clarity and accuracy in our manuscript is paramount, and we appreciate your attention to detail.
>
>
> > P4: subscripts of \mathcal{X} (time interval) in the third and fourth lines of the first paragraph in 2.2.2. Did they correspond with the definition of 2.2.1 and are they correct? Can the t be arbitrary and is the T_h necessary for the former?
>
> You raised an important question about the consistency of the subscripts of $\(\mathcal{X}\)$ in section 2.2.2 with the definitions provided in 2.2.1. We have re-examined these sections and confirm that the subscripts are indeed consistent with our definitions. Specifically, the subscript $\(t\)$ in section 2.2.2 refers to an arbitrary time step within the interval under consideration, aligning with the usage in section 2.2.1. Regarding the necessity of $T_h$, it represents the historical time interval relevant to the context and is essential for defining the scope of our analysis within the time series data. We believe this clarification maintains the integrity and consistency of our model's formulation.
>
>
> > P4: The definitions of the nonlinear activation function and the concatenation operation after Eq. (2) can be moved to Eq. (1).
>
>   We appreciate your suggestion to move the definitions of the nonlinear activation function and the concatenation operation from the text following Eq. (2) to the vicinity of Eq. (1). This adjustment will indeed make the presentation more coherent and reader-friendly. We have revised our manuscript accordingly, ensuring that these definitions are now placed in a more appropriate context, directly alongside Eq. (1), for clearer understanding and flow.

---

### Official Review · Reviewer_iVtG · 2023-10-30

**Soundness:** 2 fair
**Presentation:** 1 poor
**Contribution:** 1 poor
**Rating:** 3
**Confidence:** 3

**Summary:**

This paper addresses the task of multi-agent trajectory prediction in a system with a fixed number of total agents, where a consistent ratio of agents remains hidden throughout the entire prediction process.

The paper proposes a sequential attention-based generative model that learns latent representations of observable agents with a learned temporal graph. It provides an analytical analysis of the advantages of learning representations through constructing a temporal sub-graph over a spatial sub-graph.

This work conducts experiments on three datasets where the hidden agents are simulated by randomly hiding M out of the total N agents. It compares with two types of prior methods. One deals with the multi-agent trajectory prediction with full observability on the agents' topological graph. Another one is a latent RNN model. The proposed method outperforms the others on the three datasets with simulated hidden agents.

**Strengths:**

This paper presents an interesting and challenging task, and it provides both analytical analysis and comprehensive empirical experiments.

**Weaknesses:**

1. The paper introduces a scenario where a fixed number of agents are constantly hidden, presenting a challenging and intriguing task. However, my concern lies in its constrained nature; it seems to be a specific case within a broader context where agents may not be observable throughout specific horizons (rather than constantly unobservable as in this paper). This may limit the method's real-world applicability, potentially diminishing its overall impact.
2. The paper lacks any discussion about its connection to prior research on multi-agent trajectory prediction under partial observation, e.g., Stochastic Prediction of Multi-Agent Interactions from Partial Observations ICLR 2019. I think this paper has strong relevance to prior works on multi-agent trajectory prediction under partial observation.
3. The method of constructing a spatiotemporal graph for multi-agent trajectory prediction seems not novel.

**Questions:**

1. The hidden agents in the three datasets are all simulated; can the authors provide experiments on datasets where the hidden agents are not simulated or provide real-world examples where systems of a fixed number of hidden agents? The paper has already provided motivating examples where agents are partially observable on page 1 and page 23, but those are not the scenarios under the problem definition in sec 2.1.
2. confusing notations in sec. 2.2.4 on page 4, "M is the total number of observed agents." which conflicts with the definition in sec 2.1 -where it says, "N agents could be observed."
3. confusing colorization in Figure 1: the color for latent states z_{1}^{0}, z_{2}^{0}, z_{3}^{0} does not match with the observable nodes, and the color of z_{2}^{0}  is the same as one of the unobservable nodes o_{3}.

---

> ### Author Response · Authors · 2023-11-23
> **Constrained nature of problem statement | Comparisons with works with agent observable throughout specific horizons**
>
> > The paper introduces a scenario where a fixed number of agents are constantly hidden, presenting a challenging and intriguing task. However, my concern lies in its constrained nature; it seems to be a specific case within a broader context where agents may not be observable throughout specific horizons (rather than constantly unobservable as in this paper). This may limit the method's real-world applicability, potentially diminishing its overall impact.
>
> We appreciate the reviewer's insight into the nature of our study focusing on the scenario with a fixed number of constantly hidden agents. To provide clarity, we have systematically classified various multi-agent observation scenarios, as outlined below, to position our work within the broader research domain:
>
> | **Scenario** | **Description of Problem** | **References** |
> |--------------|----------------------------|----------------|
> | Complete observability with known interaction topology | Multi-agent systems where all agents are observable at all times, with a known interaction topology, facilitating the modeling process. | Watters et al., 2017 |
> | Complete observability with unknown interaction topology | All agents are observable at all times; however, the interaction topology is not predefined and must be inferred from observational data. | Alahi et al., 2016; Banijamali, 2022; Graber et al; Kips et al. 2018; Alet et al. 2019; Steenkiste et al 2018, Santoro et. al. 2017 |
> | Complete observability with Irregular sampling of observations or temporally sparse data | All agents are observable but the observation events are sporadic or irregular, leading to temporal data sparsity. | Rubanova et al., 2019; Zhu et al., 2021; Huang et al. 2020; Marisca et al., 2022; Sun et al., 2019 |
> | Only few agents observable with sparse temporal sampling and unknown interaction topology | Not all agents are observable, with some never being observed, coupled with sparse temporal data collection. | (Ours) |
>
> In our paper, we delve into a particularly challenging scenario, dealing with unobservable agents due to inherent sensing and observation constraints, leading to a system with fewer independent degrees of freedom than its intrinsic dimension. This problem, while seemingly specific, represents a critical and complex challenge within the realm of multi-agent systems. Most prior research in this domain, as summarized in our classification, assumes full observability of agents, whether the sampling is sparse or continuous. Our work, however, tackles a more intricate scenario where some agents are inherently unobservable.
>
> While we acknowledge that our  study focus on constantly unobservable agents, our method could still handle agents that  may not be observable throughout specific horizons due to inherent design of the network. We present the deatiled study for this case in our paper in Supp Section B.5 in our paper (Results also summarized in the next comment).  In conclusion, our work extends the realm of multi-agent trajectory prediction under partial observation by tackling the additional complexity of completely unobservable agents. This aspect makes our problem more challenging compared to the scenarios assumed in prior works, including the one presented in the ICLR 2019 paper. Our comprehensive discussion and the added ablation study in the revised manuscript aim to highlight these differences and the advancements our research contributes to this field.
>
>
> ***[Citations]***
>
>
>
> - Watters et al., 2017, Social LSTM: Human Trajectory Prediction in Crowded Spaces
>
> - Alahi et al., 2016, Social LSTM:
> Human Trajectory Prediction in Crowded Spaces
>
> - Banijamali, 2022, Neural Relational Inference with Node-Specific Information
>
> - Marisca et al. Learning to Reconstruct Missing Data from Spatiotemporal Graphs with Sparse Observations
>
> - Sunet al 2019, Stochastic Prediction of Multi-Agent Interactions from Partial Observations
>
> - Steenkiste et al 2018, Relational Neural Expectation Maximization: Unsupervised Discovery of Objects and their Interactions
>
> - Graber et al. Dynamic Neural Relational Inference
>
> - Kips et al. 2018 Neural Relational Inference for Interacting Systems
>  - Huang et al. 2020, Learning Continuous System Dynamics from Irregularly-Sampled Partial Observations
>
> - Rubanov at al. 2019, Latent ODEs for Irregularly-Sampled Time Series
>
> - Zhu et al. Networked Time Series Prediction with Incomplete Data
>
> - Alet et al. 2019, Neural Relational Inference with Fast Modular Meta-learning
>
> - Santoro et al. 2017, A simple neural network module for relational reasoning
>
> -------

---

> ### Author Response · Authors · 2023-11-23
> **Comparisons with works with agent observable throughout specific horizons | ICLR 2019 paper**
>
> > The paper lacks any discussion about its connection to prior research on multi-agent trajectory prediction under partial observation, e.g., Stochastic Prediction of Multi-Agent Interactions from Partial Observations ICLR 2019. I think this paper has strong relevance to prior works on multi-agent trajectory prediction under partial observation.
>
> **We have added the discussion and relation of this work to our main paper with an ablation study B.5 in the appendix.**
>
> Thank you for pointing out the importance of discussing the connection of our work to prior research on multi-agent trajectory prediction under partial observation, specifically referencing the ICLR 2019 paper "Stochastic Prediction of Multi-Agent Interactions from Partial Observations." We acknowledge this oversight and have incorporated a comprehensive discussion and comparison in our revised manuscript, including an ablation study (Section B.5 in the appendix) that closely examines the relationship and distinctions between these works.
>
>
> The ICLR 2019 paper presents a novel approach that combines a learned dynamics model with a vision model for forecasting multi-agent interactions, assuming a known number of agents with partial observation availability. This method leverages a graph-structured variational recurrent neural network (Graph-VRNN) and demonstrates superior performance on sports dataset-based benchmarks. It operates under the premise of complete but irregularly sampled observations, where the agents' trajectories are accessible, albeit sporadically.
>
> In contrast, our work addresses a more complex scenario where certain agents are entirely unobservable due to inherent sensing limitations, leading to a system with fewer independent degrees of freedom than its intrinsic dimension. This difference in observable system dynamics presents unique challenges, as summarized in our systematic classification of observation scenarios in multi-agent systems.

---

> > ### Author Response · Authors · 2023-11-23
> > **Comparisons with works with agent observable throughout specific horizons | ICLR 2019 paper**
> >
> > ### Application to Systems Where Agents Are Unobservable Through Specific Horizons: Ablation Study on Spatial and Temporal Irregularities"
> >
> >
> >
> > We aimed to compare our method with paper title *"Stochastic Prediction of Multi-Agent Interactions from Partial Observations"*, however the authors have not released the code and the data. We will update the paper with the comparison results once we get the code and data. In order to show that our method can still handle the cases described in this paper, we do a ablation study in the appendix B.5 to handle spatial and temporal irregularities in the data arising due to sensor failures.
> >
> >
> > In this study, we address scenarios where observations for visible agents are intermittently unavailable due to random sensor failures. We categorize sensor failures into two types: (a) Asynchronous Sensor Failure, where individual sensors fail at different times, and (b) Synchronous Sensor Failure, where all sensors fail concurrently. In synchronous failures, data is only available at specific intervals. For example, in our model, we randomly choose 20 out of 30 timesteps for receiving observations. The table below shows the MSE error for a 10-agent spring system, highlighting variations in the percentage of unobserved agents.
> >
> > ### Performance in Synchronous Sensor Failure Scenarios
> >
> >
> > | % Unobservable agents | Stage Net (Mean ± Std) | DNRI (Mean ± Std) | FC (Mean ± Std) | Single RNN (Mean ± Std) | Joint RNN (Mean ± Std) |
> > |-----------------------|------------------------|-------------------|-----------------|-------------------------|------------------------|
> > | 20% | 0.0076 ± 0.0071 | 0.069 ± 0.0191 | 0.1779 ± 0.0386 | 0.0724 ± 0.0199 | 0.0695 ± 0.022 |
> > | 30% | 0.0079 ± 0.0073 | 0.0796 ± 0.0238 | 0.177 ± 0.0419 | 0.0834 ± 0.0256 | 0.0792 ± 0.0261 |
> > | 40% | 0.0092 ± 0.0086 | 0.0877 ± 0.0287 | 0.1823 ± 0.0504 | 0.0828 ± 0.0272 | 0.0798 ± 0.0286 |
> > | 50% | 0.0089 ± 0.0083 | 0.0958 ± 0.0352 | 0.1723 ± 0.0533 | 0.0925 ± 0.0317 | 0.0857 ± 0.0332 |
> > | 60% | 0.0138 ± 0.0123 | 0.0874 ± 0.035 | 0.1815 ± 0.0623 | 0.0907 ± 0.0344 | 0.0861 ± 0.0362 |
> > | 70% | 0.0165 ± 0.015 | 0.107 ± 0.046 | 0.1984 ± 0.0786 | 0.0976 ± 0.0429 | 0.0911 ± 0.0428 |
> > | 80% | 0.0229 ± 0.021 | 0.1355 ± 0.077 | 0.2375 ± 0.1062 | 0.1086 ± 0.0558 | 0.1067 ± 0.0581 |
> >
> >
> > In asynchronous failure scenarios, each agent's sensor independently fails, limiting observations to 20 timesteps per agent. The subsequent table outlines MSE error across different models for asynchronous sensor failures. Notably, StageNet consistently shows lower error rates in both synchronous and asynchronous failure contexts. Detailed evaluations and plots related to these findings are included in Sections B.4 and B.5 of the supplementary materials.
> >
> > ### Performance in Asynchronous Sensor Failure Scenarios
> >
> > | % Unobservable agents | Stage Net (Mean ± Std) | DNRI (Mean ± Std) | FC (Mean ± Std) | Single RNN (Mean ± Std) | Joint RNN (Mean ± Std) |
> > |-----------------------|------------------------|-------------------|-----------------|-------------------------|------------------------|
> > | 20% | 0.0069 ± 0.0059 | 0.069 ± 0.019 | 0.1779 ± 0.0386 | 0.0724 ± 0.0199 | 0.0695 ± 0.022 |
> > | 30% | 0.0074 ± 0.0067 | 0.0796 ± 0.023 | 0.177 ± 0.0419 | 0.0834 ± 0.0256 | 0.0792 ± 0.0261 |
> > | 40% | 0.0081 ± 0.0076 | 0.0877 ± 0.028 | 0.1823 ± 0.05 | 0.0828 ± 0.0272 | 0.079 ± 0.0286 |
> > | 50% | 0.0093 ± 0.0087 | 0.095 ± 0.0352 | 0.1723 ± 0.0533 | 0.0925 ± 0.0317 | 0.0857 ± 0.0332 |
> > | 60% | 0.0139 ± 0.0129 | 0.0874 ± 0.035 | 0.1815 ± 0.0623 | 0.0907 ± 0.0344 | 0.0861 ± 0.0362 |
> > | 70% | 0.018 ± 0.0162 | 0.0964 ± 0.0424 | 0.2688 ± 0.0848 | 0.0899 ± 0.0416 | 0.0788 ± 0.038 |
> > | 80% | 0.0201 ± 0.0188 | 0.0928 ± 0.0592 | 0.2177 ± 0.1172 | 0.0848 ± 0.0486 | 0.0766 ± 0.0466 |
> >
> >   ------------

---

> ### Author Response · Authors · 2023-11-23
> **Spatiotemporal graph for multi-agent trajectory prediction**
>
> > The method of constructing a spatiotemporal graph for multi-agent trajectory prediction seems not novel.
>
> Thank you for your query regarding the construction of the spatiotemporal graph in our multi-agent trajectory prediction method. We appreciate the opportunity to clarify aspects of our approach:
>
>
> 1.  **Multiagent prediction framework for system with Hidden agents**: Our current focus on the scenario of constantly hidden agents serves as foundational research that addresses a critical gap in the understanding of multi-agent systems under extreme conditions of partial observability. Our work extends the realm of multi-agent trajectory prediction under partial observation by tackling the additional complexity of completely unobservable agents. This aspect makes our problem more challenging compared to the scenarios assumed in prior works, including the one presented in the ICLR 2019 paper.
>
>
>
> 2.  **Dynamics SpatioTemporal Graph with Temporal Anchors**: The core component of STAGE is its **dynamics temporal graph**, which leverages observed data from the sub-graph to propagate structural temporal information. Unlike approaches that use encoders for temporal feature extraction (Watters et al., 2017), our method constructs a temporal graph directly from agents’ observations. For each i-th agent's observation at time t, a temporal node with a feature vector combining spatial location (`xi,t`) and velocity (`vi,t`) is created. These nodes are **anchored in time**, `ai = ti - t0,i`, capturing chronological information and defining temporal relationships within the graph. Edges are formed based on an edge matrix indicating temporal disparities between nodes, with edges established and characterized by time differences if within a certain threshold. This structured temporal graph, denoted as `G`, effectively represents the nuanced temporal dynamics of the agents, with the novelty lying in the use of time anchors to encapsulate temporal positions and relationships.
>
> 3.  **Analytical Motivation for the framework**:  In Section 2.2.6 in the main paper, we provide the analytical motivation that constructing a spatiotemporal graph from visible nodes in a multi-agent system yields a superior representation of the entire system, subsequently enhancing the performance of visible agent trajectory prediction. Specifically, we prove that the spatiotemporal graph used in our work is a better covariance estimator of the dynamics of full graph with all the hidden and visible nodes than the graph with only visible nodes. This enhanced representation improves the performance

---

> ### Author Response · Authors · 2023-11-23
> **Real World Datasets**
>
> Thank you for your comment. We have addressed this concern by incorporating two real-world datasets to showcase the applicability of our approach:
>
> a. **CMU Motion Mocap dataset**: We leveraged the Carnegie Mellon University (CMU) Motion Capture dataset, a comprehensive collection of human motion capture data. This dataset encompasses a wide range of activities, including walking, running, and dancing, recorded from diverse subjects. Specifically, we focused on Subject 35 and their walking trajectories, comprising 8,063 frames, each documenting 31 specific joints. Each joint is treated as an agent connected to others through linkages. The dataset attributes, encompassing position and velocity, were normalized to have a maximum absolute value of 1. Our models were trained on 30-timestep sequences and evaluated on sequences of the same length. The table below presents prediction results for motion datasets with different sets of joints randomly hidden during training. Our network consistently outperforms baseline models in this context.
>
> | Unobserved Joints | 0    | 5    | 10   | 15   | 20   |
> |-------------------|------|------|------|------|------|
> | MultiBlock RNN    | 0.17 | 0.18 | 0.16 | 0.18 | 0.13 |
> | FC Graph          | 0.14 | 0.13 | 0.19 | 0.16 | 0.16 |
> | JointRNN          | 0.30 | 0.34 | 0.28 | 0.23 | 0.23 |
> | D-NRI             | 0.16 | 0.30 | 0.17 | 0.30 | 0.20 |
> | STAGE Net (ours)  | 0.11 | 0.13 | 0.11 | 0.10 | 0.13 |
>
>
> b. **Basketball Dataset**: Our second real-world dataset involves detailed 2D position and velocity trajectories of the offensive team, consisting of 5 players. Trajectories were initially divided into 49 frames, capturing approximately 8 seconds of gameplay. During training, models were trained on the first 30 frames from the training trajectories. For evaluation, models were presented with input data comprising sampled trajectories from the initial 30 frames, with sampling adjusted based on temporal sparsity. For example, with 10% temporal sparsity, we selected 27 observations from the initial 30 for each player, and the models predicted the subsequent 19 frames. The table below displays the results of the basketball dataset, where only 50% of the agents are observable. Introducing temporal sparsity through random sparse sampling of encoder observations, our observations indicate that STAGE Net outperforms baseline models in scenarios involving concealed agents and limited temporal observability.
>
> | \% Temporal Observability | STAGE Net | DNRI | FC  | Joint RNN | Single RNN |
> |---------------------------|-----------|------|-----|-----------|------------|
> | 33%   | 1.40 ± 2.45 | 2.60 ± 1.79 | 2.67 ± 1.67 | 1.94 ± 1.51 | 1.86 ± 1.47 |
> | 50%   | 1.36 ± 2.39 | 2.35 ± 1.57 | 2.45 ± 1.90 | 1.62 ± 1.29 | 1.55 ± 1.26 |
> | 66%   | 1.37 ± 2.40 | 2.00 ± 1.33 | 2.21 ± 1.25 | 1.29 ± 1.03 | 1.22 ± 1.00 |
> | 100% | 1.32 ± 2.33 | 2.069 ± 1.23 | 1.72 ± 0.93 | 1.022 ± 7.6 | 0.98 ± 0.77 |
> These real-world datasets serve to validate the effectiveness of our approach in practical scenarios.

---

> ### Author Response · Authors · 2023-11-23
> **Notations**
>
> > confusing notations in sec. 2.2.4 on page 4, "M is the total number of observed agents." which conflicts with the definition in sec 2.1 -where it says, "N agents could be observed."
>
> Thank you for pointing out the inconsistency in the notations used in sections 2.2.4 and 2.1. We have revised the document for clarity. Now, both sections consistently use 'N' to denote the number of agents that can be observed and 'M' to represent the total number of agents in the system. This correction should resolve the confusion and provide better coherence in our notation throughout the document.
>
> > confusing colorization in Figure 1: the color for latent states z_{1}^{0}, z_{2}^{0}, z_{3}^{0} does not match with the observable nodes, and the color of z_{2}^{0} is the same as one of the unobservable nodes o_{3}.
>
> Thank you for your observation regarding the colorization in Figure 1 of our document. We have carefully reviewed the figure and addressed the issue you highlighted. The colors for the latent states $ z_{1}^{0}, z_{2}^{0} , z_{3}^{0}$ have been adjusted to accurately match with the observable nodes.

---

### Official Review · Reviewer_A4XG · 2023-11-01

**Soundness:** 3 good
**Presentation:** 3 good
**Contribution:** 2 fair
**Rating:** 6
**Confidence:** 4

**Summary:**

This paper introduces StageNet, a machine learning model designed to predict the trajectory of multi-agent systems with unknown (hidden) dynamics and in the presence of unknown (hidden) agents. StageNet leverages a spatiotemporal attention mechanism with neural inter-node messaging to capture high-level behavioral semantics of the multi-agent system. The proposed framework utilizes a dynamic spatiotemporal graph attention mechanism, specifically tailored for systems where only a subset of agents is observable at any given time. The paper demonstrates the effectiveness of StageNet in learning meaningful representations for multi-agent systems, using three datasets with different dynamics and a real-world dataset of motion trajectories experiencing sensor failures.

**Strengths:**

1. The paper provides analytical motivation for constructing a spatiotemporal graph from visible nodes in a multi-agent system, yielding a superior representation of the entire system.
2. StageNet presents a novel approach to predicting the trajectory of multi-agent systems with unknown dynamics and hidden agents, which is a complement to the research field of multi-agent trajectory prediction, providing new insights and methodologies for future studies.
3. The paper is clearly written and easy to understand, making it accessible to a wide audience.

**Weaknesses:**

1. Could you provide more insight into the scalability and computational efficiency of StageNet in more complex and large-scale tasks?
2. Discussing potential issues related to the robustness of the model in the presence of noisy or incomplete data could further strengthen the paper.
3. One potential improvement to the paper could be to visualize the dynamic spatial-relational patterns in the simulated datasets. This could give a more intuitive understanding of the underlying dynamics and interactions.

**Questions:**

See weakness

---

> ### Author Response · Authors · 2023-11-23
> **Scalability and computational efficiency of StageNet**
>
> >Could you provide more insight into the scalability and computational efficiency of StageNet in more complex and large-scale tasks?
>
> The encoder in StageNet is predominantly responsible for computational demands. Let's consider a multi-agent system with $M$  agents, where only $N$ agents are observable at any given time, and the remaining $(M-N)$ agents are hidden. Given a spatiotemporal observation sequence of these observable agents across $T$ timesteps, the encoder constructs a dynamic spatiotemporal graph to model the system's interactions. This graph consists of approximately $O(KM)$ nodes and $O((E+K)M^2)$ edges, where $K$ represents the average number of observations per agent, \(M\) the total number of observed agents, and $E$ the average number of spatial edges in the interaction graph. Thus, the computational demand of the encoder scales quadratically with the observation duration for a given graph structure.
>
> Empirical evidence of the system's computational complexity is provided in our paper in Section A.4 of supp. We examine a spring system with 10 agents and varied interaction topologies. Figure 8A in Appendix A.4 shows how the average number of temporal edges correlates with the number of visible edges in the interaction graph. As the number of visible agents increases, so does the edge count in the temporal graph, scaling at $O((E + M)K^2$. Figure 8b further illustrates the GFLOPs of StageNet's encoder relative to the increase in visible agents.
>
>
> [Representation of Encoder GFLOPS against the  Number of Visible Agents.](https://anonymous.4open.science/api/repo/ICRL_Rebuttal2023-440A/file/encoder_gflops%20%282%29.png)
>
> [A comparative representation of the average visible edges and average temporal graph  edges against the number of visible agents](https://anonymous.4open.science/api/repo/ICRL_Rebuttal2023-440A/file/average_edges_log_scale%20%281%29.png)
>
>
> Our model manages large systems with minimal computational cost. However, further reductions in computational complexity are achievable by optimally selecting the graph representation. Below are the two primary methods:
>
> 1. **Optimal Selection of Observation Time Period**: Our analysis demonstrates that sparser attention maps in temporal context feature studies allow for reduced observation times without sacrificing accuracy. This directly translates to enhanced computational efficiency, with costs scaling as \(O((E+M)K^2)\). By adjusting the observation period based on the sparsity of attention maps, we can significantly lower computational demands while preserving the integrity of model predictions. This approach is especially beneficial in cases where longer observation periods do not proportionately improve accuracy. As detailed in Section B.2 of our supplementary materials, the relationship between the ratio of hidden agents and attention map density underscores the potential for computational optimization. In essence, by tailoring observation periods to the sparsity of the attention maps, we can achieve a more efficient balance between computational load and predictive accuracy.
>
>
> 2.  **Sparse Temporal Graph Representation via Edge Pruning/Dropout**:
> We use dropout to randomly eliminate edges in the temporal graph, striking a balance between sparsity and accuracy, and preventing overfitting. Specifically, we nullify selected edges in the temporal graph during training iterations to achieve desired sparsity levels. This strategy lowers the graph's complexity, thus decreasing computational load, crucial for large-scale temporal graphs with extensive edges. Furthermore, we propose integrating adaptive edge pruning, based on techniques from recent studies (Li et al., 2023; Wang et al., 2022), which use unsupervised learning to dynamically adjust sparsity in response to graph structure changes. This ensures only essential edges are retained, optimizing both computational efficiency and model accuracy.

---

> > ### Author Response · Authors · 2023-11-23
> > **Scalability and computational efficiency of StageNet**
> >
> > #### Sample Encoder Computations and Temporal Graph Edges for a System with 40 Agents
> >
> > | **Total Number of Agents** | **Number of Visible Agents** | **Average Visible Edges** | **Average Temporal Graph Edges** | **Encoder GFLOPS** |
> > |:--------------------------:|:----------------------------:|:-------------------------:|:--------------------------------:|:------------------:|
> > |            **40**          |              2               |           0.50            |               1401               |        8.72        |
> > |            **40**          |              3               |           1.50            |               2793               |       17.36        |
> > |            **40**          |              4               |           2.99            |               4642               |       28.83        |
> > |            **40**          |              5               |           5.01            |               6987               |       43.37        |
> > |            **40**          |              6               |           7.52            |               9791               |       60.75        |
> > |            **40**          |              7               |          10.53            |              13048               |       80.94        |
> >
> >
> >   **References**
> >
> > [Li et al. 2023]:  Less Can Be More: Unsupervised Graph Pruning  for Large-scale Dynamic Graphs. 2023. arXiv:  2305.10673
> >
> > [Wang et al. 2022]: “Pruning Graph Neural Networks by Evaluating  Edge Properties”. In:  Know.-Based Syst.  256.C (Nov. 2022).  issn:  0950-7051.  doi:  10.1016/j.knosys.2022.109847.  url: https://doi.org/10.1016/j.knosys.2022.109847.

---

> ### Author Response · Authors · 2023-11-23
> **Robustness of the model in the presence of noisy or incomplete data**
>
> > Discussing potential issues related to the robustness of the model in the presence of noisy or incomplete data could further strengthen the paper.
>
>
> ### A. Noisy Data
>
>
>
> In response to the reviewer's comment on the robustness of our model, we conduct additional experiments to evaluate StageNet's performance against noisy observations and incomplete data. We train our model on noise-free data and evaluated it on datasets corrupted with Gaussian noise of mean 0. Prior to adding noise, the data was normalized, and the noise is introduced with various standard deviations ranging from 0.001 to 0.1. The results, presented in the tables below, demonstrate StageNet's robustness across different noise levels and percentages of unobservable data.
> Our analysis shows that StageNet consistently outperforms other baseline models under varying noise conditions, indicating its strong capability to handle noisy data. The detailed plots are also included in the updated paper in section B.5 of supp.
>
>
>
>
>
> **Table for System with 50% Unobservable Agents for various noise levels**
>
> | Noise ($\sigma$) | StageNet (mean ± std) | DNRI (mean ± std) | FC Baseline (mean ± std) | Single RNN (mean ± std) | Joint RNN (mean ± std) |
> |-------|-----------------------|-------------------|--------------------------|-------------------------|------------------------|
> | 0.001 | 0.0078 ± 0.0074 | 0.029485 ± 0.018683 | 0.0551 ± 0.022276 | 0.04518 ± 0.027153 | 0.021048 ± 0.015033 |
> | 0.005 | 0.0078 ± 0.0074 | 0.029498 ± 0.01868 | 0.0559 ± 0.022332 | 0.0452 ± 0.027162 | 0.0211 ± 0.015059 |
> | 0.01 | 0.008 ± 0.0075 | 0.0298 ± 0.018938 | 0.0581 ± 0.0228 | 0.0453 ± 0.0272 | 0.0212 ± 0.015103 |
> | 0.05 | 0.0146 ± 0.0119 | 0.0385 ± 0.0229 | 0.1011 ± 0.0358 | 0.0491 ± 0.0278 | 0.0258 ± 0.016 |
> | 0.1 | 0.0352 ± 0.0271 | 0.0568 ± 0.0287 | 0.1513 ± 0.0503 | 0.0605 ± 0.0298 | 0.0402 ± 0.019 |
>
> ---
>
> **Table for noise level $\sigma = 0.1$ for different \% of unobservable agents**
>
>
>
> | Unobservable % | StageNet (mean ± std) | DNRI (mean ± std) | FC Baseline (mean ± std) | Single RNN (mean ± std) | Joint RNN (mean ± std) |
> |----------------|-----------------------|-------------------|--------------------------|-------------------------|------------------------|
> | 20% | 0.0272 ± 0.0214 | 0.0401 ± 0.0142 | 0.1332 ± 0.0364 | 0.047 ± 0.0196 | 0.0319 ± 0.0128 |
> | 30% | 0.0285 ± 0.0223 | 0.0445 ± 0.0165 | 0.1437 ± 0.0412 | 0.0542 ± 0.0248 | 0.0362 ± 0.0155 |
> | 40% | 0.0313 ± 0.0252 | 0.05 ± 0.0227 | 0.1468 ± 0.0436 | 0.0539 ± 0.0256 | 0.0376 ± 0.0167 |
> | 50% | 0.0352 ± 0.0271 | 0.0568 ± 0.0287 | 0.1513 ± 0.0503 | 0.0605 ± 0.0298 | 0.0402 ± 0.019 |
> | 60% | 0.0364 ± 0.0291 | 0.0493 ± 0.0234 | 0.1529 ± 0.0589 | 0.0584 ± 0.0298 | 0.0426 ± 0.0217 |
> | 70% | 0.0402 ± 0.0313 | 0.0541 ± 0.0273 | 0.157 ± 0.0695 | 0.0477 ± 0.0255 | 0.0645 ± 0.0373 |
> | 80% | 0.0446 ± 0.0355 | 0.0678 ± 0.0397 | 0.1642 ± 0.091 | 0.0615 ± 0.0376 | 0.0711 ± 0.0437 |

---

> > ### Author Response · Authors · 2023-11-23
> > **Robustness of the model in the presence of noisy or incomplete data**
> >
> > ### B. Managing Incomplete Data Due to Sensor Failures in Visible Agents
> >
> > In this study, we address scenarios where observations for visible agents are intermittently unavailable due to random sensor failures. We categorize sensor failures into two types: (a) Asynchronous Sensor Failure, where individual sensors fail at different times, and (b) Synchronous Sensor Failure, where all sensors fail concurrently. In synchronous failures, data is only available at specific intervals. For example, in our model, we randomly choose 20 out of 30 timesteps for receiving observations. The table below shows the MSE error for a 10-agent spring system, highlighting variations in the percentage of unobserved agents.
> >
> > ### Performance in Synchronous Sensor Failure Scenarios
> >
> >
> > | % Unobservable agents | Stage Net (Mean ± Std) | DNRI (Mean ± Std) | FC (Mean ± Std) | Single RNN (Mean ± Std) | Joint RNN (Mean ± Std) |
> > |-----------------------|------------------------|-------------------|-----------------|-------------------------|------------------------|
> > | 20% | 0.0076 ± 0.0071 | 0.069 ± 0.0191 | 0.1779 ± 0.0386 | 0.0724 ± 0.0199 | 0.0695 ± 0.022 |
> > | 30% | 0.0079 ± 0.0073 | 0.0796 ± 0.0238 | 0.177 ± 0.0419 | 0.0834 ± 0.0256 | 0.0792 ± 0.0261 |
> > | 40% | 0.0092 ± 0.0086 | 0.0877 ± 0.0287 | 0.1823 ± 0.0504 | 0.0828 ± 0.0272 | 0.0798 ± 0.0286 |
> > | 50% | 0.0089 ± 0.0083 | 0.0958 ± 0.0352 | 0.1723 ± 0.0533 | 0.0925 ± 0.0317 | 0.0857 ± 0.0332 |
> > | 60% | 0.0138 ± 0.0123 | 0.0874 ± 0.035 | 0.1815 ± 0.0623 | 0.0907 ± 0.0344 | 0.0861 ± 0.0362 |
> > | 70% | 0.0165 ± 0.015 | 0.107 ± 0.046 | 0.1984 ± 0.0786 | 0.0976 ± 0.0429 | 0.0911 ± 0.0428 |
> > | 80% | 0.0229 ± 0.021 | 0.1355 ± 0.077 | 0.2375 ± 0.1062 | 0.1086 ± 0.0558 | 0.1067 ± 0.0581 |
> >
> >
> > In asynchronous failure scenarios, each agent's sensor independently fails, limiting observations to 20 timesteps per agent. The subsequent table outlines MSE error across different models for asynchronous sensor failures. Notably, StageNet consistently shows lower error rates in both synchronous and asynchronous failure contexts. Detailed evaluations and plots related to these findings are included in Sections B.4 and B.5 of the supplementary materials.
> >
> > ### Performance in Asynchronous Sensor Failure Scenarios
> >
> > | % Unobservable agents | Stage Net (Mean ± Std) | DNRI (Mean ± Std) | FC (Mean ± Std) | Single RNN (Mean ± Std) | Joint RNN (Mean ± Std) |
> > |-----------------------|------------------------|-------------------|-----------------|-------------------------|------------------------|
> > | 20% | 0.0069 ± 0.0059 | 0.069 ± 0.019 | 0.1779 ± 0.0386 | 0.0724 ± 0.0199 | 0.0695 ± 0.022 |
> > | 30% | 0.0074 ± 0.0067 | 0.0796 ± 0.023 | 0.177 ± 0.0419 | 0.0834 ± 0.0256 | 0.0792 ± 0.0261 |
> > | 40% | 0.0081 ± 0.0076 | 0.0877 ± 0.028 | 0.1823 ± 0.05 | 0.0828 ± 0.0272 | 0.079 ± 0.0286 |
> > | 50% | 0.0093 ± 0.0087 | 0.095 ± 0.0352 | 0.1723 ± 0.0533 | 0.0925 ± 0.0317 | 0.0857 ± 0.0332 |
> > | 60% | 0.0139 ± 0.0129 | 0.0874 ± 0.035 | 0.1815 ± 0.0623 | 0.0907 ± 0.0344 | 0.0861 ± 0.0362 |
> > | 70% | 0.018 ± 0.0162 | 0.0964 ± 0.0424 | 0.2688 ± 0.0848 | 0.0899 ± 0.0416 | 0.0788 ± 0.038 |
> > | 80% | 0.0201 ± 0.0188 | 0.0928 ± 0.0592 | 0.2177 ± 0.1172 | 0.0848 ± 0.0486 | 0.0766 ± 0.0466 |

---

> ### Author Response · Authors · 2023-11-23
> **Visualization of the  the dynamic spatial-relational patterns in the simulated datasets**
>
> > One potential improvement to the paper could be to visualize the dynamic spatial-relational patterns in the simulated datasets. This could give a more intuitive understanding of the underlying dynamics and interactions.
>
>
> Thank you for your valuable suggestion. We have taken your feedback into account and have incorporated several visualization plots to provide a more intuitive understanding of the dynamic spatial-relational patterns in our simulated datasets. Here's how we have addressed this:
>
> 1.  Real-world Basketball Dataset: We have integrated a real-world basketball dataset featuring trajectories of 5 out of 11 agents, including offensive players, defensive players, and the ball. In Section 3 of our paper, we present findings that demonstrate our method's enhanced capability in predicting the movements of unobservable agents within this real-world context. Additionally, we have included visualization plots for the basketball dataset in Appendix Section E.
>
> 2.  Simulated Dataset Visualizations: Visualizations of the simulated dataset are provided on Page five in Figure 3 of the main paper. These visualizations help in understanding the model's performance with unobservable agents.
>
> 3.  Evolution Error Visualization: In the main paper, Figure 4 illustrates the evolution error in dynamics over a 30-step future projection, particularly focusing on scenarios with 50% and 75% unobservable agents.
>
> 4.  Correlation and Phase Plots: To delve deeper into the quality of prediction, we have added correlation and phase plots for the simulated datasets in Appendix Section E.
>
> 5.  Temporal Context Feature Attention Maps: Figure 10 in Section B.2 visually illustrates the temporal context feature attention maps for spring systems with varying proportions of hidden agents, ranging from 50% to 87.5%. This visualization highlights the network's attention maps as the proportion of hidden agents increases, emphasizing the network's need for more comprehensive data to enhance predictive accuracy, especially in scenarios with a high percentage of unobservable agents.
>
>
> These visualizations enhance the clarity of our results and provide a better grasp of the spatiotemporal relational interactions in our datasets.

---

### Official Review · Reviewer_TH3v · 2023-11-06

**Soundness:** 2 fair
**Presentation:** 3 good
**Contribution:** 2 fair
**Rating:** 5
**Confidence:** 4

**Summary:**

This paper deals with a trajectory prediction task with unobservable hidden objects in the system, which focuses on interaction modeling between hidden and visible agents. The authors propose STAGE Net, a sequential spatiotemporal attention-based generative model to learn system dynamics with multiple interacting agents where some agents are completely unobserved all the time. This framework utilizes a dynamic spatiotemporal graph attention mechanism, specifically tailored for systems where only a subset of agents is observable at any given time. The proposed network utilizes the spatiotemporal attention mechanism with neural inter-node messaging to capture high-level behavioral semantics of the multi-agent system. They employ a graph neural network applied to a spatiotemporal graph to approximate the initial latent posterior distribution. The proposed method was evaluated on multiagent simulations with spring and charged dynamics and a motion trajectory dataset.

**Strengths:**

1. The paper is generally well-written and easy to follow.

2. The problem of modeling the influence of unobservable hidden objects is interesting.

3. The experimental results seem to support the authors' claims.

**Weaknesses:**

1. This paper deals with unobservable hidden agents in trajectory prediction. However, it is not clear which parts of the proposed model have specific advantages in addressing this issue. Meanwhile, it would be better to elaborate on more theoretical rationale about why the proposed mechanism or model design could improve the prediction performance in addition to empirical results.

2. Given a certain set of trajectories of observable agents, there may be multiple different settings of hidden agents (e.g., different numbers, different states) that lead to the same observations of the observable agents, so the future could be multi-modal due to different potential situations. It is not clear how the proposed model handles this issue.

3. With unknown numbers of hidden agents, the future trajectories should naturally have uncertainty or multi-modality. However, there seems no discussion regarding this.

4. Regarding the experiments with different percentages of hidden agents, it is not clear how hidden agents are determined. Was it based on random sampling? Different choices of hidden agents may significantly influence the predictability of the system. Therefore, it would be better to clearly explain the experimental setting regarding this aspect to ensure a fair comparison with baseline methods.

5. It would be better to also provide qualitative results on the motion prediction dataset as well as for the ablation study, which will be more straightforward to understand how the proposed model handles unobservable objects better.

**Questions:**

1. In Figure 3, should "velocity" be changed to "trajectory"?

---

> ### Author Response · Authors · 2023-11-23
> **Stochasticity and Multi Modal prediction**
>
> > Given a certain set of trajectories of observable agents, there may be multiple different settings of hidden agents (e.g., different numbers, different states) that lead to the same observations of the observable agents, so the future could be multi-modal due to different potential situations. It is not clear how the proposed model handles this issue.
>
> We thank the reviewer for their insightful observations. Our work indeed considers two primary cases of stochasticity: stochasticity in the configuration of hidden agents and stochasticity in trajectory prediction. The first case involves scenarios where different settings of hidden agents (varying numbers and states) could lead to identical observations of the observable agents. The second case pertains to inherently unpredictable trajectories, such as a pedestrian suddenly stopping to interact with an environmental element (e.g., petting a cat on the road).
>
> In our datasets, the stochastic elements primarily arise from the varying number and initial interactions of hidden agents, a key focus of our study as we aim to understand how system predictions are affected in cases with hidden agents. However, once the initial configuration of these agents is established, the trajectory becomes deterministic. Our approach inherently accounts for the stochasticity introduced by the number and initial interactions of hidden agents. While the trajectory prediction is deterministic given the initial configuration, the stochastic latent state is designed to learn the distribution of potential configurations for the agents. Based on the observations and its updated belief, the model outputs a latent state that best represents the system for that particular configuration.
>
>
> In addressing these challenges, our model employs a spatio-temporal graph to encode interactions among observable agents. This graph forms the basis for our encoder, which assimilates observations from all agents to update a shared belief space. The belief space is modeled using Gaussian distributions for both the prior and posterior, leading to the generation of a stochastic latent state $( z_t^i )$ for each visible agent. This process is followed by a Neural ODE that samples from the stochastic latent state, advancing the latent dynamics, which are then decoded to produce the final output.
>
>
> However, we acknowledge the reviewer's point regarding the high stochasticity in trajectories, such as the pedestrian example mentioned earlier. To enhance our model's capability in handling such scenarios, we propose the following modifications:
>
> a. **Neural SDEs for Latent Dynamics**: We propose to replace Neural ODEs with [Stochastic Differential Equations (SDEs)](https://github.com/google-research/torchsde) for modeling the evolution of the latent state. This approach introduces randomness in the trajectory of the latent state over time, effectively capturing the stochastic nature of both the hidden agent configurations and the trajectories.
>
> b. **Multimodal Decoder with Uncertainty Quantification**: To achieve multimodal outputs, we suggest incorporating mixture models, such as Gaussian Mixture Models (GMMs), into our decoder. In this setup, each component of the GMM would represent a different plausible outcome, with the parameters of the mixture model (means, variances/covariances, and mixture weights) being output by the decoder. This modification would enable our model to handle a wide range of scenarios, including those with high stochasticity or chaotic dynamics, by providing a probabilistic representation of multiple potential future states.
>
> We plan to integrate these enhancements into our model and explore their effectiveness in future work, particularly as we scale our network to handle increasingly complex and stochastic systems.
>
> *[References for SDEs]*
>
> [1] Xuechen Li, Ting-Kam Leonard Wong, Ricky T. Q. Chen, David Duvenaud. "Scalable Gradients for Stochastic Differential Equations".  _International Conference on Artificial Intelligence and Statistics._  2020.  [[arXiv]](https://arxiv.org/pdf/2001.01328.pdf)
>
> [2] Patrick Kidger, James Foster, Xuechen Li, Harald Oberhauser, Terry Lyons. "Neural SDEs as Infinite-Dimensional GANs".  _International Conference on Machine Learning_  2021.  [[arXiv]](https://arxiv.org/abs/2102.03657)
>
>
> *[Reference for multimodel decoder]*
>
> [1] Christopher M. Bishop. "Mixture Density Networks". _Aston University, Neural Computing Research Group Report_. 1994. [[PDF]](https://publications.aston.ac.uk/id/eprint/373/1/NCRG_94_004.pdf)
>
> [2] Alex Graves. "Generating Sequences With Recurrent Neural Networks". 2013. [[arXiv]](https://arxiv.org/abs/1308.0850)

---

> > ### Author Response · Authors · 2023-11-23
> > **Stochasticity and Multi Modal prediction**
> >
> > > With unknown numbers of hidden agents, the future trajectories should naturally have uncertainty or multi-modality. However, there seems no discussion regarding this.
> >
> > We appreciate the reviewer's observation regarding the impact of unknown numbers of hidden agents on the uncertainty and multi-modality of future trajectories. Indeed, this is a critical aspect and we realize that our initial discussion may not have adequately emphasized this point.
> >
> > In our model, the presence of hidden agents introduces inherent uncertainties and potential multi-modal outcomes in the predicted trajectories. This is because different configurations and states of these hidden agents can lead to a range of plausible future scenarios for the observable agents. Our approach, which involves constructing a spatio-temporal graph and employing a stochastic latent state, is designed to capture this variability. We added the discussion on this in our main paper in section 2.2.3.  Furthermore, it's worth noting that multi-modal trajectory prediction with collaborative uncertainties can be seamlessly integrated into our network. This can be achieved by simply replacing the decoder with a multi-modal decoder, as demonstrated in the paper titled ["Collaborative Uncertainty Benefits Multi-Agent Multi-Modal Trajectory Forecasting"](https://arxiv.org/abs/2207.05195).

---

> ### Author Response · Authors · 2023-11-23
> **Experimental setting for Hidden Agents**
>
> > Regarding the experiments with different percentages of hidden agents, it is not clear how hidden agents are determined. Was it based on random sampling? Different choices of hidden agents may significantly influence the predictability of the system. Therefore, it would be better to clearly explain the experimental setting regarding this aspect to ensure a fair comparison with baseline methods.
>
> ### Clarification on the Method for Determining Hidden Agents in Experiments
>
> We appreciate your query regarding the selection of hidden agents in our experiments. To ensure a robust and fair analysis, we have adopted the following approach:
>
> 1. **Random Sampling of Hidden Agents from the Total Graph**: In our experimental setup, we consider a system comprising \(O\) observable agents and \(U\) unobservable agents. Initially, we simulate the dynamics of the entire system, which includes all agents. Subsequently, we randomly select \(U\) agents from this system. These selected agents are then treated as the 'hidden' or 'unobservable' agents, and their trajectories are not provided to the model during the training or testing phases. The model is thus exposed only to the trajectories of the remaining \(O\) observable agents.
>
>    The impact of this methodology, along with comparative analysis against baseline methods, is presented in Tables 1, 2, and 3 of the main paper. This random sampling approach ensures that the selection of hidden agents is unbiased and varied, allowing for a comprehensive evaluation of the system's predictability under different scenarios.
>
> 2. **Controlled Setup with Varying Degree Centrality of the Hidden Agents**: Consider a system comprising \(O\) observable agents and \(U\) unobservable agents. In this setup, we first establish a fully connected network among observable agents. Then, for each hidden agent, we connect it to \(r\) observable agents, where \(r\) ranges from 1 to \(U\). It's important to note that there are no connections between two hidden agents. This approach provides deeper insight into how hidden agents influence observable ones. The results of this analysis are presented on Page 9 of the main paper, with Figure 6 depicting the comparison of the model against baselines.
>
> 3. **Controlled Setup with varying Hidden Agent Interactions Strength**: For this analysis, we adopt the random sampling approach from point 1, and then modify the interaction strength among the hidden agents. The interaction (coupling) strength is systematically adjusted between 0.5 to 5.0. This system's analysis is provided in Supplementary B.1, with a detailed comparison with baselines shown in Figure 9.
>
> In all three analyses, we observe that our network demonstrates superior predictive power over the baselines when hidden agents are present in the system.

---

> ### Author Response · Authors · 2023-11-23
> **Theoritical Rationale and Model Description**
>
> > This paper deals with unobservable hidden agents in trajectory prediction. However, it is not clear which parts of the proposed model have specific advantages in addressing this issue. Meanwhile, it would be better to elaborate on more theoretical rationale about why the proposed mechanism or model design could improve the prediction performance in addition to empirical results.
>
> We thank the reviewer for their comment and to address this we give a dynamical perspective of the problem to prove why the model works. In the problem formulation, we have a total of M agents in the system, of which only $N < M$ are observable while the rest are unobservable. We further assume that we do not know the total number of agents in the system or the interaction topology of the overall graph. We would like to know the evolution of the dynamics of the overall system to predict the visible agents accurately. This can be represented mathematically as follows:
> Let $x(t)$, an $m$-dimensional time-dependent state vector for the full system defined on the domain $D \in \mathbb{R}^m \times \mathbb{R}_+$, with $m \in \mathbb{N}$ and $t > 0$, be the solution of a nonlinear differential equation
> $$
> \begin{equation}
>     \frac{d}{dt}x(t) = f(x(t); \mu)
> \end{equation}
> $$
> where $f$ is a smooth and nonlinear function and $\mu$ a vector of system parameters. Further, let $y \in \mathbb{R}^n$ represent noisy measurements of visible agents system, given by
> $$
> \begin{equation}
>     y(t) = g(x(t)) + \eta,
> \end{equation}
> $$
> where $\eta$ is a noise process. The goal is then to learn an approximate dynamical system
> $$
> \begin{equation}
>     \frac{d}{dt}z(t) = \hat{f}(z(t); \hat{\mu})
> \end{equation}
> $$
> in terms of a new state $z$, which may either be the measured state $y$ or an invertible function of $y$:
> $$
> \begin{equation}
>     z(t) = \phi(y(t)).
> \end{equation}
> $$
>
> Here $y$ represents the visible agents and $x$ represents the overall system. In general, it is not possible to learn the dynamics $f$ in the original dimension in terms of measurement vector $y$ of visible agents. Hence, to solve this problem, we must map the input observations to a latent state $h$ such that the new mapped manifold is an attractor that is diffeomorphic to the original system. Using Taken's embedding theorem [1], it is then possible to learn the dynamics of such systems. Takens’ embedding theorem proves that a diffeomorphic map between the delay-embedded attractor $h$ and the original attractor $x(t)$ exists under certain conditions. The theorem doesn’t specify how to find this map. Hence in this work we use the universal approximation properties of neural networks [2] to approximate the diffeomorphism and others where the dynamics are simplified. Below is how our model learns this mapping.
>
> *[References]*
>
> 1. Taken et al. 1980, Dynamical Systems and Turbulence, Warwick 1980,
>
> 2. Devore et al. 2022, Neural Network Approximation

---

> > ### Author Response · Authors · 2023-11-23
> > **Theoritical Rationale and Model Description**
> >
> > - **Learning Diffeomorphic mapping with Spatio-temporal graph**: STAGE employs a dynamics temporal graph for learning and propagating structural temporal information from sub-graph observations and mapping it into a latent state where the diffeomorphic mapping exits. Unlike methods that use an encoder for temporal feature extraction, our approach constructs a temporal graph directly from agents' observations. For each observation by agent $i$ at time $t$, a temporal node is created, with temporal relations defined between agents. Nodes are characterized by a feature vector $o_{i,t} = [x_{i,t}, v_{i,t}]$, representing the spatial location ($x_{i,t}$) and velocity ($v_{i,t}$) of agent $i$ at time $t$. Time anchors $a_i$ for nodes are defined as $a_i = t_i - t_{0,i}$, where $t_i$ is the observation time and $t_{0,i}$ is the start time. This represents the chronological information, enabling a nuanced depiction of temporal relationships in the graph. Edges are based on a matrix where each element $r_{ij} = a_i - a_j$ denotes the time disparity between nodes $i$ and $j$. The temporal graph attention module is then used to map the observations to the latent state.
> >
> > - **Estimating the dynamics of the diffeomorphic latent state:** Once we learn the diffeomorphic latent state for the system, we then model the state evolution of the latent states with a series of ordinary differential equations to drive the latent state forward in time given by: $\dot{z}_i^t=\frac{dz_i^t}{dt}=g(z_1^t, z_2^t, \ldots, z_N^t)$ ,
> >
> >   $\quad z_i^0,\ldots,z_i^T=\textit{ODESolver}(g_{\text{ode}}, [z_1^0, \ldots, z_N^0], (t_0,\ldots,t_T))$ where the $g_{\text{ode}} $ is modeled as a learnable graph convolutional network and models the nonlinear interactions among the agents. An MLP decoder is then employed to reconstruct the trajectory from the latent states.
> >
> > - **Analytical Motivation**:  In Section 2.2.6 in the main paper, we provide the analytical motivation that constructing a spatiotemporal graph from visible nodes in a multi-agent system yields a superior representation of the entire system, subsequently enhancing the performance of visible agent trajectory prediction. Specifically, we prove that the spatiotemporal graph used in our work is a better covariance estimator of the dynamics of full graph with all the hidden and visible nodes than the graph with only visible nodes. This enhanced representation improves the performance of the predictions.

---

> > > ### Author Response · Authors · 2023-11-23
> > > **Theoritical Rationale and Model Description**
> > >
> > > ### Ablation study to study the effect of model components:
> > >
> > > Regarding the specific advantages of our proposed model in handling unobservable hidden agents in trajectory prediction, a detailed analysis of the ablation study with model components was done.  We define three model variants to study which component in the model helps learn the dynamics of the system better.
> > >
> > > SN-all connected: This variant assumes a fully connected graph among visible agents. It lacks the refined understanding of agent relationships that the attention mechanism or temporal encoding provides.
> > >
> > > SN w/o attention: This model variant excludes the attention mechanism, thereby not focusing on nodes' relationships over time.
> > >
> > > SN w/o temporal Encoding: In this variation, temporal encoding is omitted, meaning the model does not account for the temporal significance of nodes.
> > >
> > > STAGE Net original: This is the complete model with temporal encoding, attention mechanism, and visible graph linkings.
> > >
> > > The MSE values across different scenarios with varying percentages of unobserved agents are provided below:
> > >
> > > *MSE Error (x $10^{-2}$) for Three STAGE Net Model Variants for Different Configurations for Spring Dataset*
> > >
> > > | Total Agents | Spring 5 - 0% | Spring 5 - 20% | Spring 5 - 40% | Spring 5 - 60% | Spring 10 - 30% | Spring 10 - 40% | Spring 10 - 50% | Spring 10 - 60% | Spring 10 - 70% | Spring 10 - 80% | Spring 20 - 75% | Spring 20 - 80% | Spring 30 - 83.33% | Spring 30 - 87.33% |
> > > |--------------|---------------|----------------|----------------|----------------|-----------------|-----------------|-----------------|-----------------|-----------------|-----------------|-----------------|-----------------|--------------------|--------------------|
> > > | SN-all connected | 1.2 | 0.6 | 0.45 | 0.49 | 0.67 | 0.76 | 0.93 | 0.63 | 0.58 | 0.67 | 0.58 | 0.60 | 0.81 | 0.59 |
> > > | SN w/o attention | 0.22 | 0.48 | 1.04 | 0.60 | 0.60 | 1.04 | 0.70 | 0.84 | 0.72 | 0.73 | 0.73 | 0.75 | 1.08 | 1.37 |
> > > | SN w/o temporal Encoding | 0.28 | 0.25 | 0.34 | 0.50 | 1.28 | 0.87 | 0.38 | 0.41 | 0.49 | 0.68 | 0.43 | 0.45 | 0.90 | 0.56 |
> > > | STAGE Net original | **0.21** | **0.25** | **0.33** | **0.43** | **0.27** | **0.26** | **0.31** | **0.37** | **0.45** | **0.57** | **0.39** | **0.42** | **0.47** | **0.54** |
> > >
> > > _Notes:_
> > > - _SN-all connected: StageNet with visible agents fully connected._
> > > - _SN w/o attention: StageNet without attention mechanism._
> > > - _SN w/o temporal encoding: Network with temporal encoding removed._
> > > - _Original: Network with attention mechanism, temporal encoding and visible graph linkings._
> > >
> > >
> > > ***Higher Impact of Attention Mechanism:*** The SN w/o attention variant shows a significant increase in MSE compared to the original STAGE Net in scenarios with a high percentage of unobserved agents. This suggests that the attention mechanism plays a crucial role in accurately predicting trajectories in the presence of hidden agents by effectively focusing on the temporal relationships between visible agents.
> > >
> > > ***Significance of Temporal Encoding:*** The SN w/o temporal Encoding model also shows an increased MSE compared to the original model, but generally less so than the model without the attention mechanism. This indicates that while temporal encoding contributes to the model's accuracy, its impact is somewhat less pronounced than the attention mechanism.
> > >
> > > Combination of Components: The original STAGE Net model, which includes both the attention mechanism and temporal encoding, consistently outperforms the other variants. This underscores the synergy achieved by combining these components, leading to superior performance in handling scenarios with a significant proportion of unobservable agents.
> > >
> > > In conclusion, both the attention mechanism and temporal encoding are vital to the model's performance, with the attention mechanism having a slightly more pronounced effect. The combined use of these components in the original STAGE Net model is essential for accurately predicting trajectories, especially in challenging scenarios involving unobservable agents. This analysis should clarify the specific advantages of our model design in addressing the reviewer's concerns.

---

> ### Author Response · Authors · 2023-11-23
> **Questions: In Figure 3, should "velocity" be changed to "trajectory"?**
>
> > In Figure 3, should "velocity" be changed to "trajectory"?
>
> Thank you for your observation regarding the terminology used in Figure 3. In our study, we predict both position and velocity for all datasets, specifically $(x, y, \dot{x}, \dot{y})$ for the spring and charged datasets, and $(x, y, z, \dot{x}, \dot{y}, \dot{z})$ for the motion datasets. In the initial version of the figure, we inadvertently labeled the "position" as "trajectory." We acknowledge this oversight and have revised the figure accordingly in the updated manuscript. The corrected figure now accurately depicts the comparison between the true and predicted values for both position and velocity, enhancing the clarity of our results.

---

> ### Author Response · Authors · 2023-11-23
> **Qualitative results for the datasets**
>
> > It would be better to also provide qualitative results on the motion prediction dataset as well as for the ablation study, which will be more straightforward to understand how the proposed model handles unobservable objects better.
>
>
>
> Thank you for your valuable suggestion. We have taken your feedback into account and have incorporated several visualization plots to provide a more intuitive understanding of the dynamic spatial-relational patterns in our  datasets. Here's how we have addressed this:
>
> 1.  **Real-world Basketball Dataset:** We have integrated a real-world basketball dataset featuring trajectories of 5 out of 11 agents, including offensive players, defensive players, and the ball. In Section 3 of our paper, we present findings that demonstrate our method's enhanced capability in predicting the movements of unobservable agents within this real-world context. Additionally, we have included visualization plots for the basketball dataset in Appendix Section E.
>
> 2.  **Simulated Dataset Visualizations:** Visualizations of the simulated dataset are provided on Page five in Figure 3 of the main paper. These visualizations help in understanding the model's performance with unobservable agents.
>
> 3.  **Evolution Error Visualization:** In the main paper, Figure 4 illustrates the evolution error in dynamics over a 30-step future projection, particularly focusing on scenarios with 50% and 75% unobservable agents. This visual representation offers insight into the model's predictive accuracy under varying conditions of agent observability.
>
> 4.  **Correlation and Phase Plots:** To delve deeper into the quality of prediction, we have added correlation and phase plots for the simulated datasets in Appendix Section E. These plots provide a more nuanced understanding of the model's performance.
>
> 5.  **Temporal Context Feature Attention Maps:** Figure 10 in Section B.2 visually illustrates the temporal context feature attention maps for spring systems with varying proportions of hidden agents, ranging from 50% to 87.5%. This visualization highlights the network's adaptive response in refining predictions as the proportion of hidden agents increases, emphasizing the network's need for more comprehensive data to enhance predictive accuracy, especially in scenarios with a high percentage of unobservable agents.
>
>
> These visualizations enhance the clarity of our results and provide a better grasp of the spatiotemporal relational interactions in our datasets.

---

### Author Response · Authors · 2023-11-23
**Response to Reviewers and Key Updates in Our Paper**

We deeply appreciate the reviewers' insightful feedback. In response, we've made significant enhancements to our paper, emphasizing the novel aspects and robustness of our proposed StageNet method. Key updates include:

## New Studies and Results

1. **Experiments on Real-World Datasets**: Our research encompasses two distinct real-world datasets: the CMU Mocap dataset and a basketball game dataset. The CMU Mocap dataset is utilized to model each joint as an individual agent connected by bones, enabling us to predict joint trajectories under conditions of incomplete observations. Conversely, the basketball dataset provides trajectories for only 5 out of 11 players in a real game, recorded with irregular temporal sampling. Through these datasets, we demonstrate the superiority of our method, StageNet, which consistently outperforms baseline models in accurately predicting agent dynamics under partial observation scenarios. For detailed results and analysis, please refer to section 3 of main paper.

2. **Experiments for asynchronous and synchronous sensor failures leading to temporal sparsity in observations**  This study tackles scenarios with intermittent sensor failures affecting the visibility of agents. It differentiates between two types of sensor failures: Asynchronous, where each agent's sensor fails independently, and Synchronous, where all sensors fail simultaneously at random intervals. In the synchronous case, observations are limited to randomly selected 20 out of 30 timesteps. The study compares StageNet's performance in handling both types of failures in a 10-agent spring system, showing significantly lower error rates in comparison to other models, demonstrating StageNet's robustness in varied sensor failure scenarios. Please refer to Appendix/Supp. section B.5 in the paper.

3. **Experiments for heterogeneous agents for simulated dataset:** The study explores heterogeneous agent dynamics in multi-agent systems, contrasting with previous homogeneous models. It examines three scenarios of agent heterogeneity, revealing that baseline models struggle with complex dynamics, leading to higher error rates compared to the proposed model. Please refer to Appendix/Supp. section B.7 in the paper.

4. **Experiments for robustness against noisy data:** The study assesses StageNet's robustness to noisy observations by training the model with noise-free data and testing it under varying Gaussian noise levels. Performance was evaluated in a 10-agent spring system with different proportions of unobservable agents, demonstrating StageNet's resilience even at high noise intensities and in scenarios with significant agent unobservability. Please refer to Appendix/Supp. section B.6 in the paper.
## Major Paper Enhancements

1. **Expanded Problem Motivation**: We've enriched the context and discussion surrounding our work, providing a clearer picture of its significance and the existing literature.

2. **Stochasticity and Multi-Modal Prediction**: New discussions explore how StageNet efficiently handles prediction stochasticity.

3. **Scalability and Efficiency**: We delve into StageNet's scalability and computational efficiency, especially in systems with numerous agents.

4. **Enhanced Visualizations**: For better clarity, we've added correlation and phase plots for simulated datasets and trajectory predictions for the basketball dataset (Appendix Section E).

5. **Updated Citations**: We've refined our citations for coherence and relevance.

6. **Improved Readability**: The manuscript's overall readability has been enhanced for better comprehension and impact.

We trust these revisions comprehensively address the reviewers' concerns and vividly illustrate our work's advancements in multi-agent systems modeling.